# Normalized Attention Guidance:
# Universal Negative Guidance for Diffusion Models

**Dar-Yen Chen**[1,2]  **Hmrishav Bandyopadhyay**[1]  **Kai Zou**[2]  **Yi-Zhe Song**[1]

[1]SketchX, CVSSP, University of Surrey    [2]NetMind.AI
{d.chen, h.bandyopadhyay, y.song}@surrey.ac.uk  kz@netmind.ai

https://chendaryen.github.io/NAG.github.io

## Abstract

Negative guidance – explicitly suppressing unwanted attributes – remains a fundamental challenge in diffusion models, particularly in few-step sampling regimes. While Classifier-Free Guidance (CFG) works well in standard settings, it fails under aggressive sampling step compression due to divergent predictions between positive and negative branches. We present Normalized Attention Guidance (NAG), an efficient, training-free mechanism that applies extrapolation in attention space with L1-based normalization and refinement. NAG restores effective negative guidance where CFG collapses while maintaining fidelity. Unlike existing approaches, NAG generalizes across architectures (UNet, DiT), sampling regimes (few-step, multi-step), and modalities (image, video), functioning as a *universal* plug-in with minimal computational overhead. Through extensive experimentation, we demonstrate consistent improvements in text alignment (CLIP Score), fidelity (FID, PFID), and human-perceived quality (ImageReward). Our ablation studies validate each design component, while user studies confirm significant preference for NAG-guided outputs. As a model-agnostic inference-time approach requiring no retraining, NAG provides effortless negative guidance for all modern diffusion frameworks – pseudocode in the Appendix!

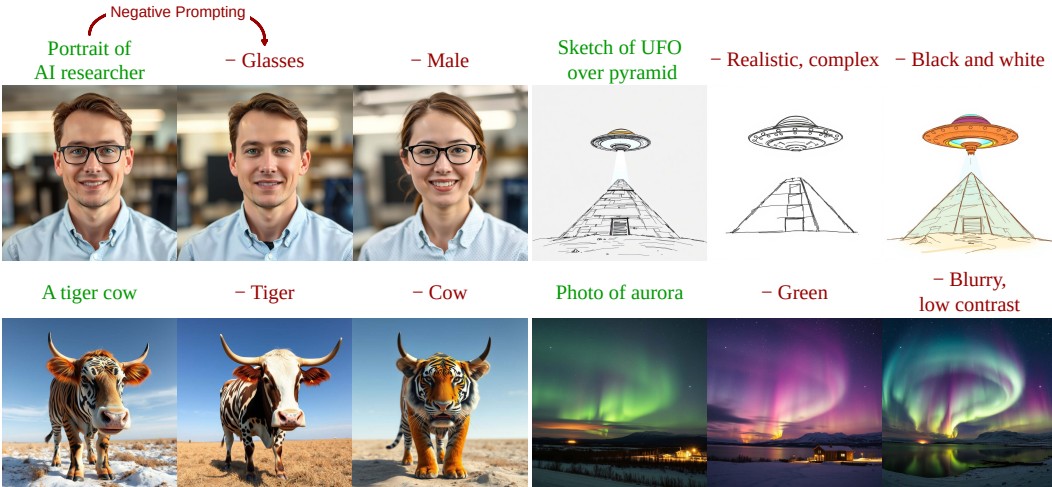

Figure 1: **Negative prompting on 4-step Flux-Schnell [1].** CFG fails in few-step models. NAG restores effective negative prompting, enabling direct suppression of visual, semantic, and stylistic attributes, such as "glasses," "tiger," "realistic," or "blurry." This enhances controllability and expands creative freedom across composition, style, and quality—including prompt-based debiasing.

39th Conference on Neural Information Processing Systems (NeurIPS 2025).

# 1 Introduction

Diffusion models have revolutionized visual synthesis, setting new standards for photorealism in image [2–4] and video [5, 6] generation. Despite advances in quality and efficiency, a critical limitation persists: effective negative guidance – suppressing unwanted attributes – particularly in few-step sampling regimes [7,8]. This capability is crucial for content safety, quality control, creative expression, and debiasing applications in real-world deployments.

The prevailing approach to diffusion model control, Classifier-Free Guidance (CFG) [9], enables negative guidance by extrapolating between positive and negative conditional outputs at each denoising step.[1] However, in few-step regimes, CFG's assumption of consistent structure between diffusion branches breaks down, as these branches diverge dramatically at early steps. This divergence causes severe artifacts rather than controlled guidance, precisely when negative guidance is most needed for high-efficiency inference scenarios.

Although several recent CFG variants [10–12] mitigate specific limitations, they inherit the same fundamental constraint: reliance on structural similarity between extrapolated outputs. Attempts to circumvent this limitation, such as NASA [13], modify attention activations directly but lack proper constraints, potentially leading to out-of-manifold feature drift, instability and feature collapse – particularly in modern DiT architectures [1, 14, 15] where feature spaces are more complex and sensitive to perturbation.

In this paper, we propose *Normalized Attention Guidance* (NAG), a simple and effective approach to negative guidance that works universally across sampling regimes, model architectures, and generation domains with minimal overhead. The core insight of NAG is to apply extrapolation directly in attention feature space, complemented by L1-norm-based normalization and feature refinement to constrain feature magnitude while preserving directional guidance. This enables effective suppression of undesired attributes without compromising fidelity across diverse settings and generation tasks.

Methodologically, NAG operates by computing attention outputs from both positive and negative prompts, then performing controlled extrapolation in this feature space: $\widetilde{Z} = Z^+ + \phi \cdot (Z^+ - Z^-)$. Unlike previous approaches that directly impose guidance on model predictions, this maintains semantic coherence even when conditional branches diverge significantly. Intuitively, NAG computes a vector away from undesired attributes (e.g., "glasses", "blurry") and moves attention features along this direction, while constraining them to avoid straying from meaningful representations. This is achieved through two key stabilization mechanisms: (1) L1-norm-based feature normalization that acts as a "guardrail", preserving directional information, and (2) feature refinement that pulls extreme features back toward familiar territory through interpolation. As detailed in Section 4.2, this creates a structured trajectory through attention space, preventing the instability inherent to direct subtraction approaches like NASA [13].

The universality of NAG is demonstrated across three dimensions: (i) *Sampling regimes* – NAG works in few-step models (1-8 steps) where CFG fails and in multi-step models alongside existing methods; (ii) *Model architectures* – NAG functions across UNet and DiT architectures without architecture-specific adjustments; (iii) *Generation domains* – NAG extends to both image and video, improving alignment and temporal coherence. Ablation studies confirm each component's contribution and the method's robustness across hyperparameters. NAG requires no retraining and integrates as a simple plug-in to existing pipelines. One immediate benefit? A simple change lets AI researchers break free from the stereotypical representation! (Figure 1)

Our contributions are as follows: (i) We introduce NAG, a universal, training-free attention guidance method that provides stable, controllable negative guidance across the diffusion model ecosystem. (ii) We restore effective negative guidance in few-step diffusion models where traditional CFG fails completely, while also enhancing negative control in multi-step diffusion when integrated with existing guidance methods. (iii) We demonstrate consistent improvements across diverse architectures (UNet, DiT), sampling regimes (1-25+ steps), and metrics (CLIP Score [16], FID [17], PFID [7], ImageReward [18]). (iv) We validate NAG's generalization to video diffusion without domain-specific modifications, improving both semantic alignment and motion characteristics through effective negative guidance.

---

[1] CFG serves as both a powerful alignment mechanism for text-to-image generation and an essential tool for negative guidance in real-world applications.

## 2 Related Works

**Diffusion models.** Diffusion models [3, 19] form the foundation of modern generative modeling, driving advances in image [2, 4, 20, 21] and video synthesis [5, 22–25]. Early methods rely on stochastic differential equations (SDEs) [26, 27] to learn a reverse denoising process from noise to data. Recently, deterministic formulations based on ordinary differential equations (ODEs) [28–34], such as Rectified Flow [29] and Flow Matching [35], have emerged as efficient alternatives, learning continuous trajectories that transform noise into data. These methods accelerate convergence and improve stability, especially in large-scale models. In parallel, architectural innovation has shifted from UNet backbones [3] to Diffusion Transformers (DiT) [1, 6, 14, 15, 36–41], offering greater scalability [14, 42, 43].

To improve inference efficiency in large-scale models, recent efforts compress the sampling trajectory to 1-8 steps [7, 8, 29, 44–54]. These few-step models encompass both UNet and DiT families, facilitating low-latency generation. However, aggressive step reduction disables classifier-free guidance (CFG) [13, 50], limiting control. Our work bridges this gap with a training-free attention-space method that restores controllability in few-step models.

**Sampling guidance.** Sampling guidance directs generation toward target semantics while improving fidelity. Classifier Guidance [19] uses classifier gradients, but requires auxiliary models and additional training. Classifier-Free Guidance (CFG) [9] avoids this by interpolating between conditional and unconditional predictions. Several advancements [10–12, 55, 56] address CFG's limitations, such as fidelity degradation and inaccurate estimation [57]. Parallel works modify attention mechanism to guide generation without negative conditioning. Self-Attention Guidance (SAG) [58] and Perturbed Attention Guidance (PAG) [59] work on self-attention layers, offering enhanced sample structure. PLADIS [60] focus on cross-attention layers to improve text alignment. However, most approaches assume semantic alignment across branches, an assumption that breaks under few-step sampling. In few-step sampling, conditional and unconditional predictions diverge, making extrapolation unreliable and causing collapse [13]. NASA [13] attempts to mitigate this by adjusting cross-attention features directly, but suffers from instability due to the feature out-of-manifold issue, especially in DiT models.

We analyze attention feature manipulation and study how to stabilize this process via extrapolation, normalization, and refinement. This leads to a robust and generalizable guidance mechanism that operates consistently across UNet and DiT architectures, and across both few-step and multi-step diffusion sampling.

## 3 Background

We briefly review the fundamentals of diffusion models for text-to-image generation; background on flow-based models is provided in Appendix B.

### 3.1 Text-to-Image Diffusion Models

Diffusion models [3] synthesize images by iteratively refining noisy samples through a learned reverse process. Training involves simulating a forward Markovian process in which clean images $x_0$ are progressively perturbed by adding Gaussian noise $\epsilon \sim \mathcal{N}(0, \mathbf{I})$ over $T$ discrete timesteps, producing intermediate noisy states $x_t$.

$$x_t = \sqrt{\bar{\alpha}_t} x_0 + \sqrt{1 - \bar{\alpha}_t} \epsilon, \quad t \in \{1, \ldots, T\}, \tag{1}$$

where $\bar{\alpha}_t$ corresponds to coefficients from a noising schedule, controlling the signal-to-noise ratio (SNR) [3, 26] across timesteps. $x_0$ is reconstructed through training a neural network $G_\theta$ to estimate the added noise $\epsilon$ as $\hat{\epsilon} = G_\theta(x_t)$ and plugging it back into Eq. (1):

$$x_0 = \frac{x_t - \sqrt{1 - \bar{\alpha}_t} \hat{\epsilon}}{\sqrt{\bar{\alpha}_t}}. \tag{2}$$

In text-to-image tasks, the model estimates noise as $\hat{\epsilon} = G_\theta(x_t, c)$, where $c$ represents text conditioning. Cross-attention units [2] integrate text embeddings into the reverse process, enabling semantic alignment between prompts and generated images.

### 3.2 Classifier-Free Guidance

Classifier guidance [19] enhances diffusion models by using gradients from a pretrained classifier to steer generation toward a target class. Classifier-Free Guidance (CFG) [9] removes the classifier dependency by extrapolating between predictions conditioned on negative and positive prompts.

Given a noisy sample $x_t$, the model predicts noise $\hat{\epsilon}^+ = G_\theta(x_t, c^+)$ using the positive condition $c^+$ and $\hat{\epsilon}^- = G_\theta(x_t, c^-)$ for the negative condition. CFG then applies guidance by extrapolating beyond the positive prediction:

$$\hat{\epsilon}^{\text{CFG}} = \hat{\epsilon}^+ + \phi \cdot (\hat{\epsilon}^+ - \hat{\epsilon}^-), \tag{3}$$

where $\phi \geq 0$ is the guidance scale controlling the trade-off between fidelity and diversity. At inference, the estimated noise $\hat{\epsilon}^{\text{CFG}}$ replaces $\hat{\epsilon}$ in Eq. (2), biasing generation toward the positive condition, and away from the negative condition.

### 3.3 Negative-Away Steer Attention

NASA [13] introduces attention-space guidance by directly modifying cross-attention outputs. For image query $Q$ and text key-value pairs $(K, V)$, attention is generally computed as:

$$Z = \text{Softmax}\left(\frac{QK^\top}{\sqrt{d_k}}\right) V. \tag{4}$$

To steer generation, NASA independently obtains $Z^+$ and $Z^-$ using $(K^+, V^+)$ and $(K^-, V^-)$ from positive and negative prompts, then subtracts scaled $Z^-$ to guide attention:

$$Z^{\text{NASA}} = Z^+ - \phi \cdot Z^-. \tag{5}$$

This enables directional control without manipulating estimated noise.

## 4 Methodology

CFG [9] applies an extrapolation on noise predictions, which can be rewritten in terms of $x_0$ by substituting Eq. (2) into Eq. (3):

$$x_0^{\text{CFG}} = x_0^+ + \phi \cdot (x_0^+ - x_0^-), \tag{6}$$

where $x_0^+$ and $x_0^-$ are reconstructed from positive and negative conditions, respectively. CFG inherently assumes a multi-step denoising process to keep these branches aligned, where samples are gradually refined across iterations. In few-step models, however, $x_0^+$ and $x_0^-$ diverge significantly after a single inference step, making the extrapolation fundamentally ill-posed. As shown on the left of Figure 3, directly applying CFG introduces severe artifacts. NASA [13] sidesteps output-space extrapolation by modifying cross-attention features. However, the absence of constraints leads to out-of-manifold shifts and degraded synthesis quality (Figure 2). We observe that this instability is further amplified in DiT architectures, motivating the need for a more stable guidance strategy.

| DiT, Flux–Schnell, 4 steps

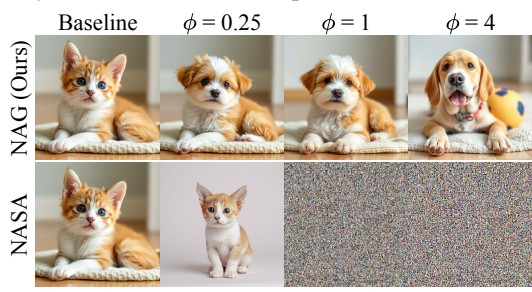

A pet.  – Cat.

| UNet, NitroSD–Realism, 2 steps

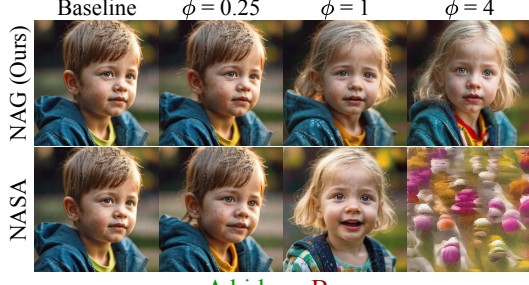

A kid.  – Boy.

Figure 2: **Comparison of NAG against NASA.**

### 4.1 First-Order Analysis of Guidance

To analyze why CFG [9] degrades under few-step sampling, we use a first-order Taylor view. Let one denoising update at time $t$ be defined by

$$z: \ c \mapsto G_\theta(x_t, c).$$

A Taylor expansion around $c$ yields

$$z(c + \Delta c) = \sum_{n=0}^{\infty} \frac{1}{n!} (\Delta c^\top \nabla_c)^n z(c) = z(c) + (\Delta c^\top \nabla_c) z(c) + \tfrac{1}{2}(\Delta c^\top \nabla_c)^2 z(c) + \cdots. \tag{7}$$

Keeping only first-order terms gives the output-space finite difference

$$\Delta z \triangleq z(c + \Delta c) - z(c) \approx \nabla_c z(c) \, \Delta c. \tag{8}$$

This relation underlies CFG's output-space extrapolation, by treating $\Delta z$ as a first-order estimate of the gradient with respect to $c$.

In multi-step sampling each denoising step is small. When $z(c + \Delta c)$ remains within a smooth locally linear neighbourhood of $z(c)$, the higher-order terms in Equation (7) are negligible and the single-shot estimator $\Delta z$ is reliable. In few-step regimes the required denoising is large and $z$ varies sharply with $c$ which makes higher-order terms non-negligible and causes CFG to break down.

Although the end-to-end mapping $c \mapsto z(c)$ is not locally linear, individual layers often operate more linearly. We factor the network at an intermediate layer $l$ into a prefix $G_\theta^l$ that produces features and a tail $\bar{G}_\theta^l$ that maps those features to the output

$$F_l(c) = G_\theta^l(x_t, c), \qquad z(c) = \bar{G}_\theta^l(F_l(c), c), \tag{9}$$

where $F_l(c)$ is the feature of layer $l$. By the chain rule, we have

$$\frac{\partial z}{\partial c} = J_{\bar{G}_\theta^l}(F_l(c)) \frac{\partial F_l}{\partial c} + \frac{\partial \bar{G}_\theta^l}{\partial c}, \tag{10}$$

where $J$ denotes the Jacobian. By repeatedly selecting an intermediate layer $l'$ of $\bar{G}_\theta^l$ and applying the chain rule to $\frac{\partial \bar{G}_\theta^l}{\partial c}$, we obtain the layer-summed form

$$\frac{\partial z}{\partial c} = \sum_l J_{\bar{G}_\theta^l}(F_l(c)) \frac{\partial F_l}{\partial c}. \tag{11}$$

Applying a first-order approximation in all $l$, we define

$$\Delta F_l \triangleq F_l(c + \Delta c) - F_l(c) \approx \frac{\partial F_l}{\partial c}(c) \Delta c. \tag{12}$$

This gives the corresponding output increment via feature steps

$$\Delta z_{\text{feature}} \triangleq \sum_l J_{\bar{G}_\theta^l}(F_l) \Delta F_l \approx \nabla_c z(c) \Delta c, \tag{13}$$

which provides an alternative first-order update: a sum of small, layer-wise edits propagated by Jacobians.

## 4.2 Normalized Attention Guidance

*Normalized Attention Guidance* (NAG) addresses instability in attention-based guidance by applying extrapolation in attention feature space, followed by Normalization and Refinement. Each component (*i.e.* extrapolation, normalization and refinement) constrains out-of-manifold shifts while preserving the semantic direction of guidance (Figure 3).

**Extrapolation in attention space.** Motivated by the first-order feature-space estimator in Equation (13), we move CFG-style extrapolation from output space to attention features and apply

$$\widetilde{Z} = Z^+ + \phi(Z^+ - Z^-), \tag{14}$$

where $\phi$ is the guidance scale and $Z^+, Z^-$ are features under positive and negative conditions. This is the feature-space analogue of CFG and it distributes guidance across layers. In few-step regimes editing features tends to stay closer to local linearity than editing the output directly. Large $\phi$ can still drive features off the manifold, so the following normalization and refinement stages bound the guidance and maintain stability.

**L1-Based Normalization.** To constrain the extrapolated features, we compute the point-wise L1 norm ratio between $\widetilde{Z}$ and $Z^+$:

$$R[i] = \frac{\|\widetilde{Z}[i]\|_1}{\|Z^+[i]\|_1}, \quad i \in \{1, \ldots, l\}, \tag{15}$$

where $Z \in \mathbb{R}^{l \times d}$ is the attention output of sequence length $l$ and dimension $d$. We then apply clipping and rescaling:

$$\widehat{Z}[i] = \frac{\min(R[i], \tau)}{R} \cdot \widetilde{Z}[i], \quad i \in \{1, \ldots, l\}. \tag{16}$$

The threshold $\tau$ limits feature magnitude, preventing extreme activations while maintaining directionality. We adopt the L1 norm to normalize attention outputs, as it preserves low-magnitude activations that often encode subtle semantics. Since attention features undergo linear projections, even small components can significantly influence the final representation. Unlike L2 or max norms, which disproportionately shrink these values, L1 retains essential structure for precise guidance.

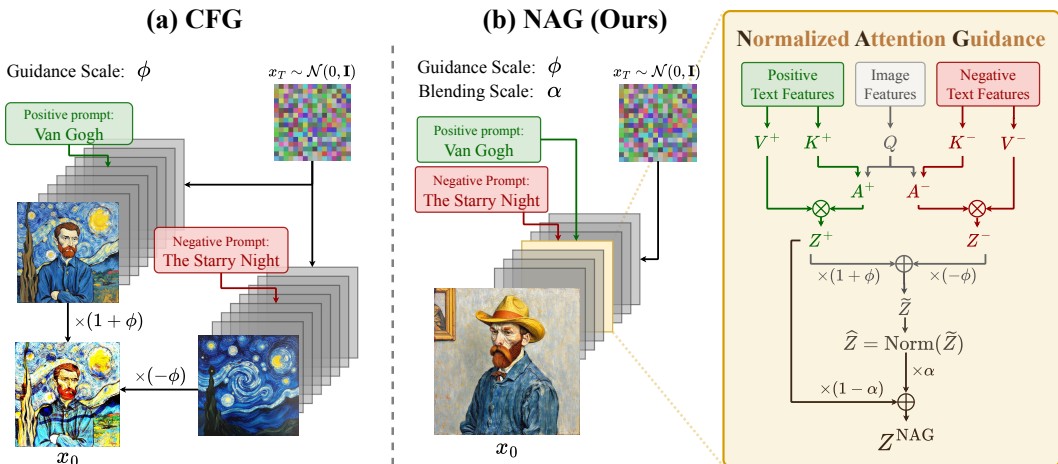

Figure 3: **Comparison of CFG and NAG in single-step sampling.** *Left:* Classifier-Free Guidance (CFG) [9] generates $x_0^+$ and $x_0^-$ from positive and negative prompts, then applies output-space extrapolation. In few-step models, $x_0^+$ and $x_0^-$ differ significantly due to coarse denoising, leading to severe artifacts rather than controlled guidance. *Right:* Normalized Attention Guidance (NAG) operates in attention space by extrapolating positive and negative features $Z^+$ and $Z^-$, followed by L1-based normalization and $\alpha$-blending. This constrains feature deviation, suppresses out-of-manifold drift, and achieves stable, controllable guidance.

**Feature Refinement.** Though normalization constrains magnitude, it may still disrupt alignment with the original distribution. To mitigate this, we blend $\widehat{Z}$ with the positive baseline:

$$Z^{\text{NAG}} = \alpha \cdot \widehat{Z} + (1 - \alpha) \cdot Z^+. \qquad (17)$$

This blending serves as a regularizer, pulling features toward the stable manifold $Z^+$ and ensuring that the guidance remains bounded in both magnitude and semantics.

**Geometric interpretation.** Figure 4 visualizes the trajectory of guided features in NAG. Raw extrapolation produces $\widetilde{Z}$, which may extend beyond the distribution of valid features. Normalization produces $\widehat{Z}$ by scaling $\widetilde{Z}$ into a bounded region (Guidance Boundary), mitigating out-of-manifold risk. The final refinement produces $Z^{\text{NAG}}$ by blending $\widehat{Z}$ with $Z^+$, contracting the feature into the Refinement Manifold. This progressive regularization preserves directional guidance from $Z^-$ while maintaining distributional consistency with $Z^+$.

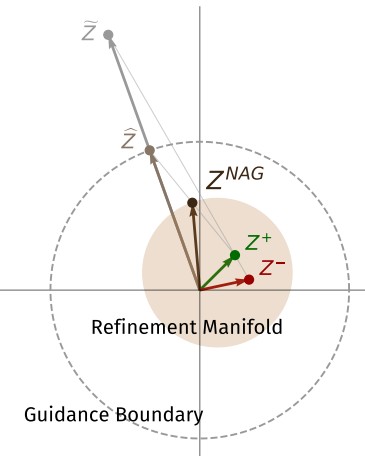

Figure 4: **Visualization of NAG.**

## 5 Experiments

Unless otherwise specified, experiments are conducted on Flux-Schnell [1] with 4-step sampling, using an NVIDIA A100 GPU. We evaluate NAG on the COCO-5K dataset [61] using CLIP Score [16], Fréchet Inception Distance (FID) [17], Patch FID (PFID) [7], and ImageReward [18], which reflects human aesthetic preference. Following NASA [13], we use "Low resolution, blurry" as a universal negative prompt for quantitative comparison. Implementation and default settings are detailed in Appendix C.

### 5.1 Evaluating NAG in Few-Step Sampling

Few-step diffusion models facilitate rapid inference, but generally lack support for CFG, making negative guidance ineffective. We evaluate NAG on both DiT-based SANA-Sprint [53], Flux-Schnell [1], SD3.5-Large-Turbo [15, 52], and UNet-based NitroSD-Realism [8], DMD2-SDXL [50], SDXL-Lightning [7]. We also include 25-step Flux-Dev [1], which lacks CFG support. Figure 1 and Figure 5 show that NAG enables guidance across concepts, ranging from removing undesired

Table 1: **Quantitative results of NAG.**

| Arch | Model | Steps | CLIP (↑) | | FID (↓) | | PFID (↓) | | ImageReward (↑) | |
|---|---|---|---|---|---|---|---|---|---|---|
| | | | Base. | NAG | Base. | NAG | Base. | NAG | Base. | NAG |
| DiT | SANA-Sprint | 2 | 31.4 | **31.9** (+0.5) | 30.29 | **28.31** (−1.98) | 37.56 | **33.29** (−4.27) | 1.008 | **1.075** (+0.067) |
| | Flux-Schnell | 4 | 31.4 | **32.0** (+0.6) | 25.47 | **24.46** (−1.01) | 38.26 | **34.95** (−3.31) | 1.029 | **1.099** (+0.070) |
| | SD3.5-Large-Turbo | 8 | 31.4 | **31.8** (+0.4) | 29.97 | **29.81** (−0.18) | 44.37 | **41.87** (−2.50) | 0.944 | **1.118** (+0.174) |
| | Flux-Dev | 25 | 30.9 | **31.5** (+0.6) | 31.04 | **28.11** (−2.93) | 43.22 | **39.01** (−4.21) | 1.066 | **1.166** (+0.100) |
| UNet | NitroSD-Realism | 1 | 31.8 | **32.4** (+0.6) | 26.21 | **23.98** (−2.23) | 30.53 | **28.73** (−1.80) | 0.847 | **0.948** (+0.101) |
| | DMD2-SDXL | 4 | 31.6 | **32.2** (+0.6) | 24.79 | **23.32** (−1.47) | 27.11 | **25.61** (−1.50) | 0.876 | **0.960** (+0.084) |
| | SDXL-Lightning | 8 | 31.1 | **31.8** (+0.7) | 27.01 | **24.99** (−2.02) | 34.02 | **31.70** (−2.32) | 0.730 | **0.842** (+0.112) |

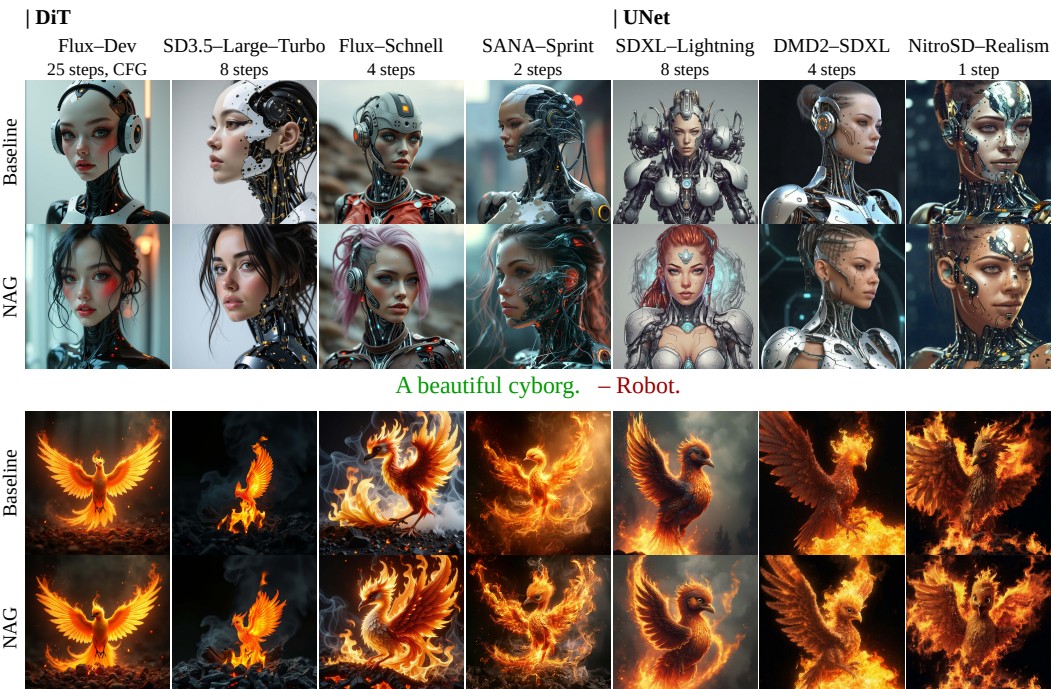

Figure 5: **Qualitative results of NAG.** NAG enhances controllability in models lacking CFG, improving semantic alignment and visual quality across architectures and sampling steps.

objects to refining perceptual quality. In the *Cyborg* vs. *Robot* example, NAG attenuates robotic features like metallic textures, producing more human-like depictions. In the *Phoenix* sample, using the universal negative prompt "Low resolution, blurry" results in enhanced sharpness and gradient contrast, yielding more vivid flame structures.

Table 1 reports consistent and significant improvements across all models and metrics. CLIP gains indicate enhanced prompt alignment, while FID and PFID reductions reflect improved fidelity. Notably, ImageReward increases across the board, validating perceptual gains through aesthetic preference metrics. These results show that NAG restores effective negative prompting in few-step models without retraining, achieving guidance previously unachievable in this regime.

## 5.2 Comparison with NASA

We compare NAG against NASA [13] in Figure 2. On DiT-based Flux-Schnell [1], NASA exhibits strong instability even at low scale, producing unnatural textures and broken images. On UNet-based NitroSD-Realism [8], NASA performs reasonably at low $\phi$ but degrades as scale increases, introducing distortions. In contrast, NAG consistently maintains output quality while enforcing negative constraints. Across architectures, it enables stronger guidance without compromising fidelity at high scale. This reflects the effectiveness of normalized feature-space extrapolation, and highlights NAG's stability, generality in regimes where prior methods collapse.

Table 2: **Quantitative results of NAG with CFG and PAG.**

| Arch | Model | Steps | Setting | CLIP (↑) | | FID (↓) | | PFID (↓) | | ImageReward (↑) | |
|---|---|---|---|---|---|---|---|---|---|---|---|
| | | | | w/o NAG | NAG | w/o NAG | NAG | w/o NAG | NAG | w/o NAG | NAG |
| DiT | SD3.5-Large | 25 | CFG | 31.8 | **32.0** (+0.2) | **25.07** | 25.42 (+0.35) | 31.68 | **31.63** (−0.05) | 1.029 | **1.130** (+0.101) |
| | | | CFG + PAG | 31.5 | **31.8** (+0.3) | 24.49 | **24.35** (−0.14) | **37.93** | 39.09 (+1.16) | 0.939 | **1.063** (+0.124) |
| UNet | SDXL | 25 | CFG | 31.9 | **32.7** (+0.8) | 23.25 | **20.90** (−2.35) | 30.01 | **27.90** (−2.11) | 0.791 | **0.906** (+0.115) |
| | | | CFG + PAG | 31.5 | **32.3** (+0.8) | 26.25 | **23.53** (−2.72) | 35.58 | **31.80** (−3.78) | 0.748 | **0.914** (+0.166) |

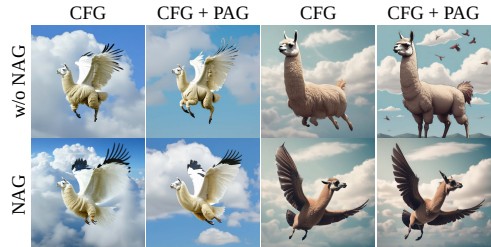

An anthropomorphic cat thoughtfully paints an oil self-portrait on canvas...
– Human face, low resolution, blurry.

An origami fox running in the forest. The fox is made of polygons...
– Static, low resolution, blurry.

Figure 6: **Qualitative video results for Wan2.1-T2V-14B.**

## 5.3 Integrating NAG with Other Guidance

To evaluate compatibility with existing guidance, we integrate NAG into 25-step SD3.5-Large [15] and SDXL [62] models alongside CFG [9] and PAG [12] in Figure 7. In the *llama-bird hybrid* case, NAG disambiguates concepts, guiding generation toward the intended bird anatomy. In the *astronaut* case, it enhances clarity and accentuates prompt-relevant attributes without distorting composition. Results in Table 2 confirm that NAG complements existing methods: Although FID and PFID vary, CLIP scores and ImageReward consistently improve. These results demonstrate that NAG is a general enhancement to standard guidance strategies, offering advancements in multi-step models.

## 5.4 Video Diffusion Models

To validate cross-modal generality, we apply NAG to the video model Wan2.1-T2V-14B [6] for generating 5-second, 480p videos. Figure 6 demonstrates that NAG enables effective negative prompting in video generation. In the first example, NAG suppresses unintended facial features and enhances detail in the fur and back-

| DiT, SD3.5-Large, 25 steps | UNet, SDXL, 25 steps

CFG · CFG + PAG · CFG · CFG + PAG

w/o NAG · NAG

A llama-bird hybrid creature flying in the sky
Llama head, bird body.
– Llama body.

w/o NAG · NAG

A tiny astronaut hatching from an egg on the moon.
– Low resolution, blurry.

Figure 7: **NAG integration with CFG and PAG.**

ground. In the second example, including "static" in the negative prompt produces outputs with stronger motion dynamics. These results show that NAG extends robust control to video synthesis, enabling content suppression and quality enhancement. It highlights NAG's capacity as a general-purpose mechanism for diffusion control across spatial and temporal domains.

## 5.5 Computational Cost

Unlike CFG [9], which requires doubling the computation of sampling steps, NAG only applies additional computation to cross-attention layers or MM-DiT [15] blocks. Table 3 reports the per-step latency across families. In Flux [1], NAG incurs a similar cost to CFG, whereas in SD3.5-Large [15], SANA [63], SDXL [62] and Wan2.1 [6], it introduces significantly lower additional inference time.

Table 3: **Per-step sampling latency.**

| Model Family | Baseline | CFG | NAG |
|---|---|---|---|
| Flux | 487ms | +488ms (+100%) | +426ms (**+87%**) |
| SD3.5-Large | 231ms | +219ms (+95%) | +109ms (**+43%**) |
| SANA | 39ms | +35ms (+90%) | +5ms (**+13%**) |
| SDXL | 75ms | +25ms (+34%) | +17ms (**+22%**) |
| Wan2.1 | 10.7s | +10.7s (+100%) | +1.3s (**+12%**) |

## 5.6 User Study

We conduct a user study to assess perceived improvements in three aspects: text alignment, visual appeal, and for video, motion dynamics. Text alignment measures how well outputs match the prompt semantics. Visual appeal reflects aesthetic quality,

Table 4: **User preferences study.**

| Model | Modal | Steps | CFG | Text | Visual | Motion |
|---|---|---|---|---|---|---|
| Flux-Schnell | Image | 4 | ✗ | +25.0% | +33.9% | – |
| SD3.5-Large | Image | 25 | ✓ | +9.2% | +15.5% | – |
| Wan2.1-14B | Video | 25 | ✓ | +20.5% | +8.7% | +14.3% |

and motion dynamics captures temporal coherence and realism in video. Details are provided in Appendix H. We compare generations using NAG against the same models without NAG in Table 4. Following Lin *et al.* [54], the preference score is computed as $(P - N)/(P + S + N)$, where $P$, $N$, and $S$ denote preferred, non-preferred, and similar votes. A score of 0% indicates equal preference, while +100% and -100% signify complete preference for or against NAG, respectively.

On few-step Flux-Schnell [1], which lacks CFG, users clearly prefer NAG for both text alignment and visual quality. For SD3.5-Large [15] with CFG, improvements are smaller but still consistent. For the video model Wan2.1-14B [6], NAG achieves notable preference in text relevance and motion quality. Overall, NAG improves user-perceived quality across models and modalities.

## 5.7 Ablation Study

**Effectiveness of components.** To assess the contribution of each design choice, we perform an ablation by removing the refinement and normalization operations. As shown on the left of Figure 8, full NAG maintains stable performance across scales, achieving continuously improving CLIP as well as optimal FID and ImageReward. Omitting refinement and normalization causes rapid degradation beyond $\phi > 5$, confirming their necessity. Visual results on the right of Figure 8 support the quantitative findings. Full NAG preserves structure and color at high guidance strength ($\phi = 20$), suppressing undesired concepts without sacrificing fidelity. Without refinement, mild distortion appears at $\phi = 5$, increases at $\phi = 10$, and results in failure at $\phi = 20$. Without normalization, artifacts emerge early and intensify with higher guidance, due to unbounded scaling. These results confirm that normalization and refinement are both critical for stable and effective negative guidance.

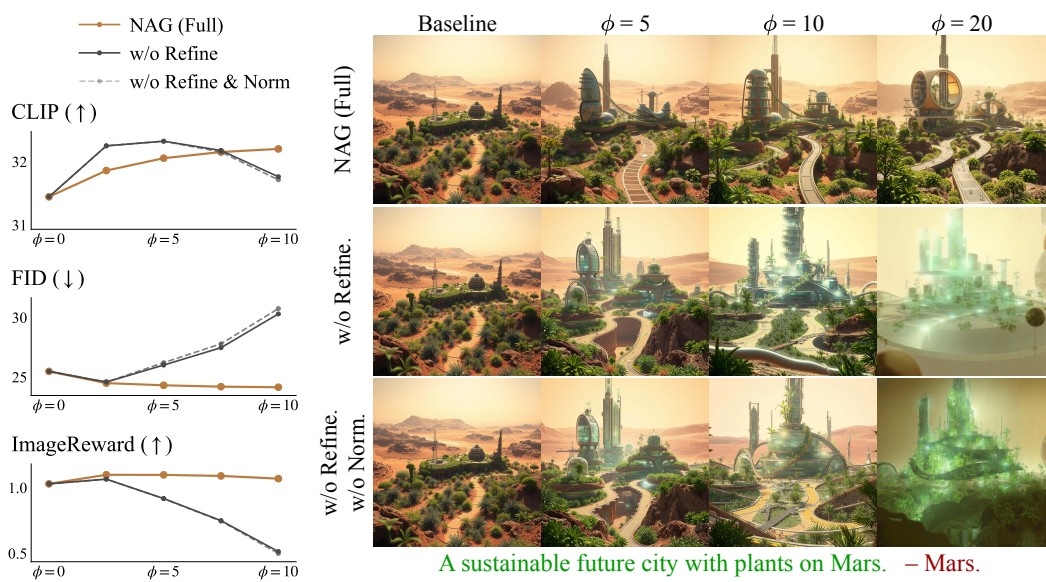

Figure 8: *Left:* Ablation study of NAG. ***Right:*** Visual results from ablation study.

**Impact of guidance scale.** We evaluate NAG's performance under varying guidance scales in Figure 9. As scale increases, negative prompt influence strengthens: in the *woman* example, the figure is gradually removed, isolating the flowing dress; in the *wolf* example, higher scales reveal sharper structure and more fine-grained details, until excessive guidance introduces distortions and saturation issues. These trends illustrate both the strength and limits of NAG: moderate scales enable refined control, while extreme scales risk artifacts or unnatural stylization.

Figure 10 further analyzes performance across different inference steps. Notably, CLIP increases monotonically, while FID and ImageReward peak at moderate scales (4–8), indicating tradeoffs

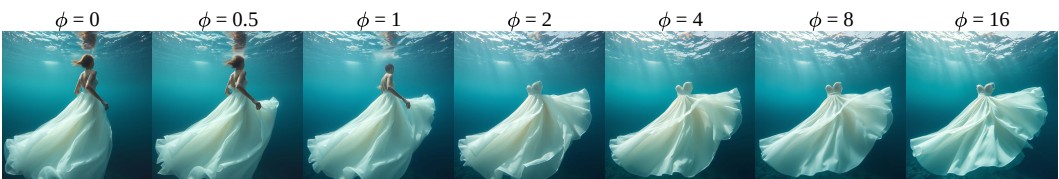

$\phi = 0$    $\phi = 0.5$    $\phi = 1$    $\phi = 2$    $\phi = 4$    $\phi = 8$    $\phi = 16$

An elegant dress fluttering under the sea.    – Woman.

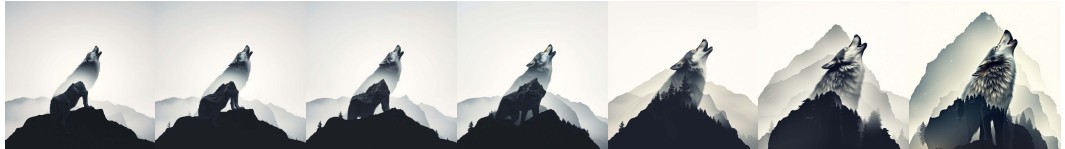

A majestic wolf howling inside the silhouette of a mountain,
double exposure photography style, surreal and dreamlike.    – Low resolution, blurry.

Figure 9: **Impact of NAG scale**.

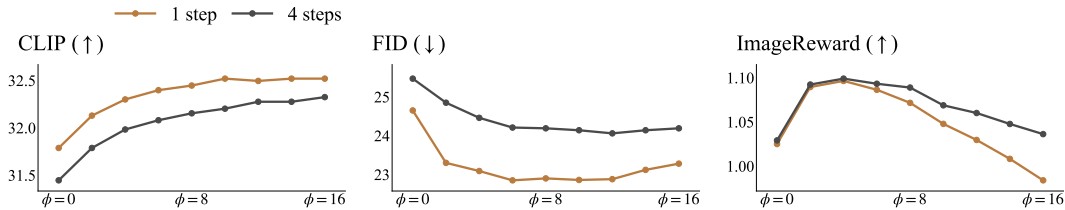

Figure 10: **Quantitative comparison of NAG scale.**

between alignment and quality. Sampling with more steps tolerates higher scales before severe degradation, allowing stronger guidance without compromising stability.

# 6 Limitations and Future Work

While NAG exhibits effective negative guidance, some failure cases remain. As shown in Figure 11, it may struggle to fully suppress certain concepts. Despite extrapolation is constrained,

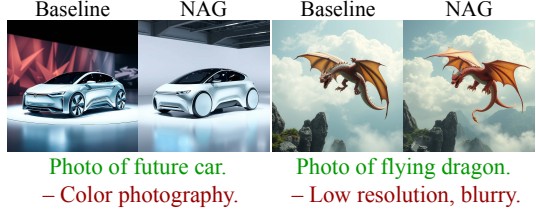

Photo of future car.    Photo of flying dragon.
– Color photography.    – Low resolution, blurry.

Figure 11: **Examples of NAG failure cases.**

excessive guidance or poorly formulated prompts may still trigger instability or texture collapse. Future work may explore finer-grained interventions within the attention mechanism to improve sensitivity to stylistic cues and suppress global artifacts. Adaptive attention weighting or token-wise modulation could improve responsiveness to nuanced or under-specified negative prompts. Additionally, refining feature calibration or incorporating local structural priors may offer more precise control over semantic and stylistic attributes.

# 7 Conclusion

We present Normalized Attention Guidance (NAG), a training-free mechanism that restores and enhances controllability in diffusion models via stable attention-space guidance. Unlike existing approaches, NAG addresses the the limitations of negative guidance in few-step models, where output-space extrapolation methods like CFG [9] fails due to divergent predictions. By applying extrapolation directly in attention features and stabilizing it with L1-based normalization and refinement, NAG mitigates out-of-manifold shifts while preserving guidance direction. NAG generalizes across architectures (UNet [3], DiT [14]), sampling strategies (few-step, multi-step), and generation domains (image, video), serving as a universal inference-time plug-in. It integrates seamlessly with CFG and PAG [12], enhancing controllability without retraining. Extensive experiments demonstrate consistent improvements in text alignment, visual fidelity, and human-perceived quality. These findings establish NAG as a simple yet powerful solution for universal control in modern diffusion models – negative prompting that actually works!

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

# Table of Contents

## A   Broader Impact

By improving the effectiveness of negative prompting, NAG's flexibility enables positive impact across creative fields, including graphic design, film production, and educational content, through more reliable visual generation in both image and video domains. However, the same controllability can be misused. NAG could potentially facilitate more precise deepfake creation, amplification of harmful biases, or content manipulation by making it easier to suppress or alter specific visual traits. Its training-free nature reduces the barrier for adapting it to sensitive or malicious applications.

We strongly advocate for responsible usage and recommend that future work explore safeguards such as output detection mechanisms. While NAG expands the practical utility of diffusion models, its ethical deployment remains essential.

## B   Flow-Based Models

Flow-based generative models establish a continuous transformation between a base distribution and the data distribution using a time-dependent trainable velocity field $v_\theta(x_t, c)$. Given a pair of data points $(x_0, x_1) \sim (p_{\text{base}}, p_{\text{data}})$, intermediate states are constructed by linear interpolation:

$$x_t = (1-t)x_0 + tx_1, \quad t \in [0, 1], \tag{18}$$

where $x_0$ is sampled from a known distribution such as $\mathcal{N}(0, \mathbf{I})$, and $x_1$ is a real data sample.

At inference time, a new sample is generated by solving the ordinary differential equation:

$$\frac{dx_t}{dt} = v_\theta(x_t, c), \tag{19}$$

which moves the initial point $x_0$ along a learned trajectory toward the data distribution as $t \to 1$.

# C  Implementation Details

We provide the default NAG hyperparameters for different model families in Table 5. Algorithm 1 provides PyTorch-style pseudocode for integrating NAG into cross-attention layers.

Table 5: **Default NAG hyperparameters.**

| Architecture | Model Family | $\phi$ | $\tau$ | $\alpha$ |
|---|---|---|---|---|
| DiT | Flux | 4 | 2.5 | 0.25 |
| | SD3.5-Large | 4 | 2.5 | 0.125 |
| | SANA | 4 | 2.5 | 0.375 |
| | PixArt-$\Sigma$ | 4 | 2.5 | 0.375 |
| | Wan2.1 | 4 | 2.5 | 0.25 |
| UNet | SDXL | 2 | 2.5 | 0.5 |
| | Playground | 2 | 2.5 | 0.5 |
| | SD1.5 | 2 | 2.5 | 0.375 |

---

**Algorithm 1** Cross-Attention with NAG.

---

```python
class NAGCrossAttnProcessor:
    def __init__(self, nag_scale=2.0, tau=2.5, alpha=0.5):
        self.nag_scale = nag_scale
        self.tau = tau
        self.alpha = alpha

    def __call__(
        self, attn, image_emb, text_emb_positive, text_emb_negative,
    ):
        query = attn.to_q(image_emb)

        # Compute key K and value V for both positive and negative prompt
        key_positive = attn.to_k(text_emb_positive)
        value_positive = attn.to_v(text_emb_positive)
        key_negative = attn.to_k(text_emb_negative)
        value_negative = attn.to_v(text_emb_negative)

        # Compute attention output Z for both positive and negative prompt
        z_positive = F.scaled_dot_product_attention(
            query, key_positive, value_positive,
        )
        z_negative = F.scaled_dot_product_attention(
            query, key_negative, value_negative,
        )

        # Equation 7
        z_tilde = z_positive + self.nag_scale * (z_positive - z_negative)

        # Equation 8
        norm_positive = torch.norm(z_positive, p=1, dim=-1, keepdim=True)
        norm_tilde = torch.norm(z_tilde, p=1, dim=-1, keepdim=True)
        ratio = norm_tilde / norm_positive

        # Equation 9
        z_hat = torch.where(ratio > self.tau, tau, ratio) / ratio * z_tilde

        # Equation 10
        z_nag = self.alpha * z_hat + (1 - self.alpha) * z_positive

        return hidden_states
```

---

# D Extended Evaluation

We expand our evaluation to encompass additional diffusion models. For few-step generation, we include Hyper-Flux-Dev [48], Flux-Turbo-Alpha [64], and Hyper-SDXL [48]. For standard sampling with CFG and PAG, we test SANA 1.6B [63], PixArt-Σ [37], SD1.5 [2], and Playground v2.5 [65].

Quantitative results in Table 6 and qualitative comparisons in Figure 12 further validate the effectiveness and generality of NAG across diverse architectures and sampling regimes.

Table 6: **Additional quantitative results of NAG.**

| Arch | Model | Steps | CFG/PAG | CLIP (↑) | | FID (↓) | | PFID (↓) | | ImgReward (↑) | |
|---|---|---|---|---|---|---|---|---|---|---|---|
| | | | | Base. | NAG | Base. | NAG | Base. | NAG | Base. | NAG |
| DiT | Hyper-Flux-Dev | 8 | ✗ | 31.7 | **32.3** (+0.6) | 24.65 | **23.08** (−1.57) | 28.12 | **25.51** (−2.61) | 1.025 | **1.096** (+0.071) |
| | Flux-Turbo-Alpha | 8 | ✗ | 31.2 | **31.8** (+0.6) | 29.53 | **27.28** (−2.25) | 36.14 | **33.74** (−2.40) | 0.899 | **1.012** (+0.113) |
| | SANA 1.6B | 25 | CFG+PAG | 31.4 | **31.8** (+0.4) | 34.79 | **32.74** (−2.05) | 51.56 | **48.64** (−2.92) | 0.933 | **1.138** (+0.205) |
| | PixArt-Σ | 25 | CFG | 31.4 | **31.7** (+0.3) | 31.01 | **30.36** (−0.65) | 42.27 | **40.50** (−1.77) | 0.920 | **1.024** (+0.104) |
| UNet | Hyper-SDXL | 8 | ✗ | 31.5 | **32.0** (+0.5) | 32.36 | **31.08** (−1.28) | 39.90 | **38.11** (−1.79) | 0.931 | **1.021** (+0.090) |
| | SD1.5 | 25 | CFG | 31.2 | **31.7** (+0.5) | **23.04** | 23.18 (+0.14) | **25.19** | 26.48 (+1.29) | 0.191 | **0.337** (+0.146) |
| | Playground v2.5 | 25 | CFG | 31.5 | **32.0** (+0.5) | 34.80 | **30.45** (−4.35) | 46.74 | **46.48** (−0.26) | 1.162 | **1.163** (+0.001) |

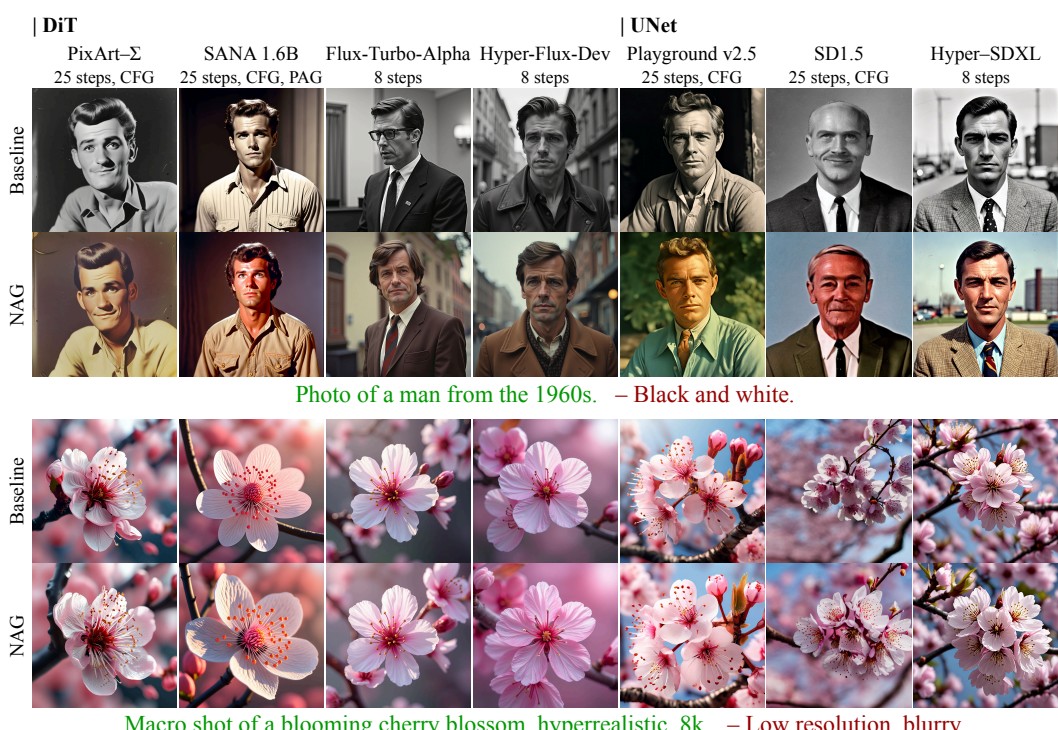

| DiT                                                          | UNet
PixArt–Σ | SANA 1.6B | Flux-Turbo-Alpha | Hyper-Flux-Dev | Playground v2.5 | SD1.5 | Hyper–SDXL
25 steps, CFG | 25 steps, CFG, PAG | 8 steps | 8 steps | 25 steps, CFG | 25 steps, CFG | 8 steps

Photo of a man from the 1960s. – Black and white.

Macro shot of a blooming cherry blossom, hyperrealistic, 8k. – Low resolution, blurry.

Figure 12: **Additional qualitative results of NAG.**

# E Extended Comparison with NASA

We quantitatively compare NAG against NASA [13] in Table 7. Since NASA fails with DiT architecture, we focus on UNet-based models: NitroSD-Realism [8], DMD2-SDXL [50], and SDXL-Lightning [7]. Across all metrics, NAG consistently outperforms NASA, demonstrating superior text alignment (CLIP), fidelity (FID, PFID), and human-perceived quality (ImageReward).

Furthermore, we conduct a user study for direct evaluation of perceptual quality. To guarantee a thorough assessment, we expand the original six positive-negative prompt pairs from Nguyen *et al.* [13] to sixteen pairs, as detailed in Appendix J.2. For text alignment, participants are instructed to consider both the positive prompt, which specifies the desired content, and the negative prompt, which defines the attribute to be suppressed. As summarized in Table 8, users exhibit a strong preference for NAG-guided outputs, reflecting improvements in both text alignment and visual appeal. The results demonstrate effectiveness of NAG in practical scenarios.

Table 7: **Quantitative results of NAG against NASA.**

| Arch | Model | Steps | CLIP (↑) | | FID (↓) | | PFID (↓) | | ImageReward (↑) | |
|---|---|---|---|---|---|---|---|---|---|---|
| | | | NASA | NAG | NASA | NAG | NASA | NAG | NASA | NAG |
| UNet | NitroSD-Realism | 1 | 32.0 | **32.4** (+0.4) | 25.86 | **23.98** (−1.88) | 30.01 | **28.73** (−1.28) | 0.900 | **0.948** (+0.048) |
| | DMD2-SDXL | 4 | 31.9 | **32.2** (+0.3) | 24.44 | **23.32** (−1.12) | 26.74 | **25.61** (−1.13) | 0.915 | **0.960** (+0.045) |
| | SDXL-Lightning | 8 | 31.6 | **31.8** (+0.2) | 26.38 | **24.99** (−1.39) | 32.39 | **31.70** (−0.69) | 0.805 | **0.842** (+0.037) |

Table 8: **User preferences study of NAG against NASA.**

| Model | Steps | Text Alignment | Visual Appeal |
|---|---|---|---|
| DMD2-SDXL | 4 | +46.0% | +56.8% |

# F Ablation Study on $\tau$ and $\alpha$

Besides the guidance scale $s$, NAG introduces two additional hyperparameters: the normalization threshold $\tau$ and the refinement factor $\alpha$. While our default values generalize well across model families and use cases, this section investigates how each affects generation behavior. $\tau$ controls the upper bound of the normalized attention magnitude. On the left of Figure 13, lower $\tau$ overly restrict guidance strength, while higher values may introduce instability and artifacts—especially under extreme guidance scales. In practice, a moderate threshold achieves a balanced trade-off between stability and effectiveness. $\alpha$ regulates how much of the guidance is retained. As depicted in the right of Figure 13, lower $\alpha$ improves robustness by maintaining closeness to the original positive features, but may weaken guidance. With $\alpha=0.125$, the model fails to fully suppress the red tone specified in the negative prompt. Larger values enable stronger guidance, but risk of collapse at high $s$.

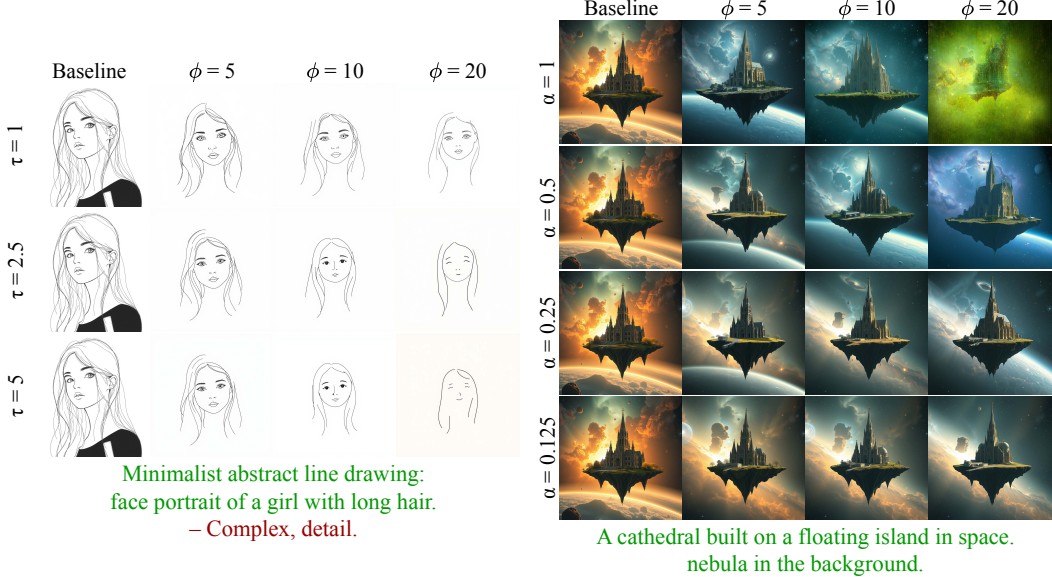

Figure 13: *Left:* Impact of $\tau$. *Right:* Impact of $\alpha$.

# G  Early Stopping of NAG

To understand the impact of NAG across timesteps, we visualize intermediate outputs during the sampling process in Figure 14. The modification from negative guidance is most prominent at early stages, with diminishing influence as denoising proceeds.

Motivated by this, we investigate early stopping of NAG. Specifically, we define a threshold $\theta \leq 1$ and apply NAG only to the first $\theta$ portion of the denoising steps. Figure 15 presents qualitative results for different $\theta$. Even with $\theta = 0.25$, outputs remain competitive. For example, in right columns, where the negative prompt targets global structure (e.g., close-up), all settings produce similar results. In contrast, higher $\theta$ improves details for fine-grained edits like the smoke representation of lion's

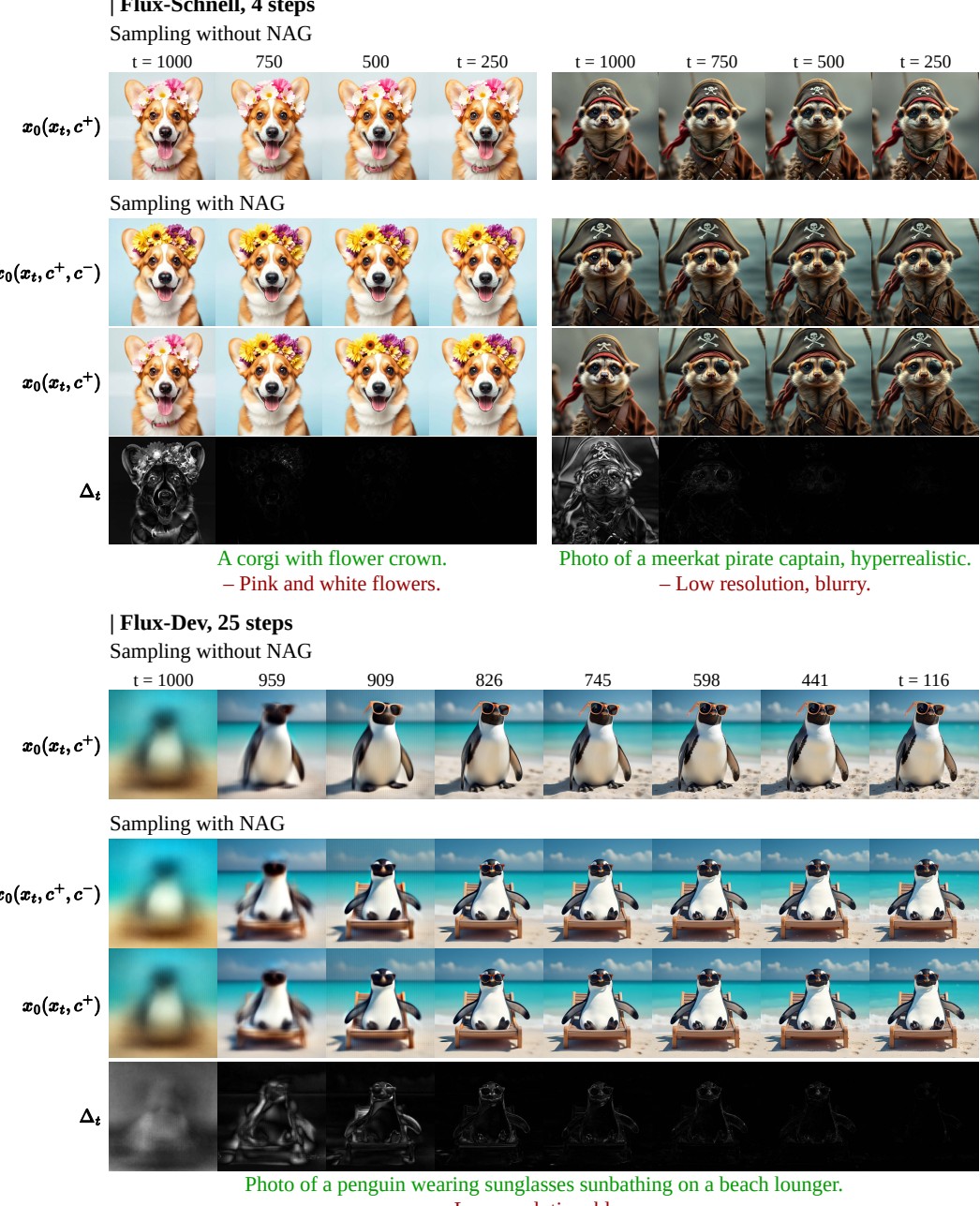

Figure 14: **Visualization of the sampling process with NAG.**

face. Quantitative results in Table 9 support this. On few-step models like Flux-Schnell [1] and DMD2-SDXL [50], $\theta = 0.25$ performs comparably to the full application with $\theta = 1.0$, significantly reducing inference time. On 25-step sampling such as PixArt-$\Sigma$ [37] and SDXL [62], lower $\theta$ slightly reduces performance, but results remain strong.

In summary, early stopping of NAG preserves most benefits while reducing computation. For various use cases, limiting NAG application to the initial steps represents a practical balance between quality and computational efficiency.

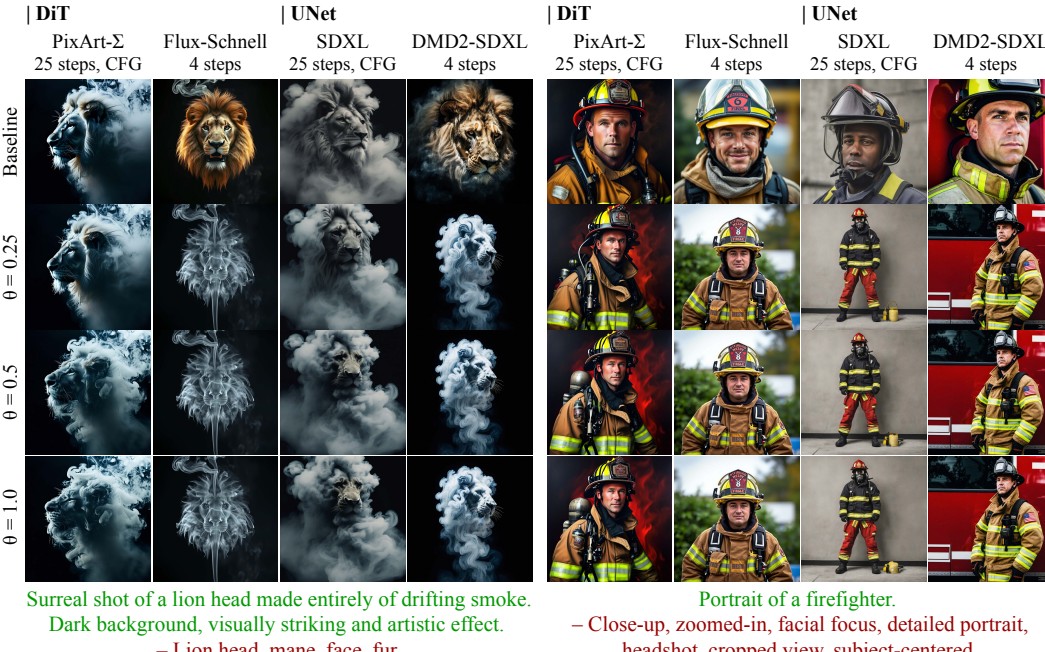

Surreal shot of a lion head made entirely of drifting smoke.
Dark background, visually striking and artistic effect.
– Lion head, mane, face, fur.

Portrait of a firefighter.
– Close-up, zoomed-in, facial focus, detailed portrait,
headshot, cropped view, subject-centered.

Figure 15: **Qualitative results of early stopping.**

Table 9: **Quantitative comparison of early stopping.**

| Architecture | Model | Steps | CFG | $\theta$ | CLIP (↑) | FID (↓) | PFID (↓) | Image Reward (↑) | Latency (↓) |
|---|---|---|---|---|---|---|---|---|---|
| DiT | Flux-Schnell | 4 | ✗ | 0 | 31.4 | 25.47 | 38.26 | 1.029 | 2.11s |
| | | | | 0.25 | **32.0** (+0.6) | **24.44** (-1.03) | **34.94** (-3.32) | 1.098 (+0.069) | 2.95s (+40%) |
| | | | | 0.5 | **32.0** (+0.6) | 24.49 (-0.98) | 34.98 (-3.27) | **1.100** (+0.071) | 3.36s (+59%) |
| | | | | 1.0 | **32.0** (+0.6) | 24.46 (-1.01) | 34.95 (-3.31) | 1.099 (+0.070) | 3.75s (+78%) |
| | PixArt-$\Sigma$ | 25 | ✓ | 0 | 31.4 | 31.01 | 42.27 | 0.920 | 2.78s |
| | | | | 0.25 | 31.6 (+0.2) | **30.25** (-0.76) | 41.00 (-1.27) | 0.965 (+0.045) | 2.89s (+4%) |
| | | | | 0.5 | **31.7** (+0.3) | 30.31 (-0.70) | **40.49** (-1.78) | 1.013 (+0.093) | 2.97s (+7%) |
| | | | | 1 | **31.7** (+0.3) | 30.36 (-0.65) | 40.50 (-1.77) | **1.024** (+0.104) | 3.10s (+12%) |
| UNet | DMD2-SDXL | 4 | ✗ | ✗ | 31.6 | 24.79 | 27.11 | 0.876 | 0.53s |
| | | | | 0.25 | 32.1 (+0.5) | 23.40 (-1.39) | 25.53 (-1.58) | 0.950 (+0.074) | 0.60s (+14%) |
| | | | | 0.5 | **32.2** (+0.6) | 23.40 (-1.39) | **25.40** (-2.71) | 0.958 (+0.082) | 0.62s (+17%) |
| | | | | 1 | **32.2** (+0.6) | **23.32** (-1.47) | 25.61 (-1.50) | **0.960** (+0.084) | 0.66s (+25%) |
| | SDXL | 25 | ✓ | ✗ | 31.9 | 23.25 | 30.01 | 0.791 | 2.72s |
| | | | | 0.25 | 32.5 (+0.6) | 21.79 (-1.46) | 28.92 (-1.09) | 0.888 (+0.097) | 2.81s (+3%) |
| | | | | 0.5 | 32.6 (+0.7) | 21.17 (-2.08) | 28.39 (-1.62) | **0.908** (+0.117) | 2.86s (+5%) |
| | | | | 1.0 | **32.7** (+0.8) | **20.90** (-2.35) | **27.90** (-2.11) | 0.906 (+0.115) | 2.97s (+9%) |

# H   User Study Details

To perform the human preference study for the text-to-image task, we randomly sample 100 prompts from the Pick-a-Pic v2 test dataset [66], excluding sensitive content. For text-to-video synthesis, we select 50 prompts from the Wan2.1 community. All generated samples undergo manual screening to ensure safety to present no risks to participants.

Each comparison involves two samples: one generated with NAG and the other without NAG. Participants were asked to select the sample with superior quality according to three criteria: (1) alignment with the prompt, (2) visual appeal, and (3) natural motion dynamics (for video). The user study interface is displayed in Figure 16.

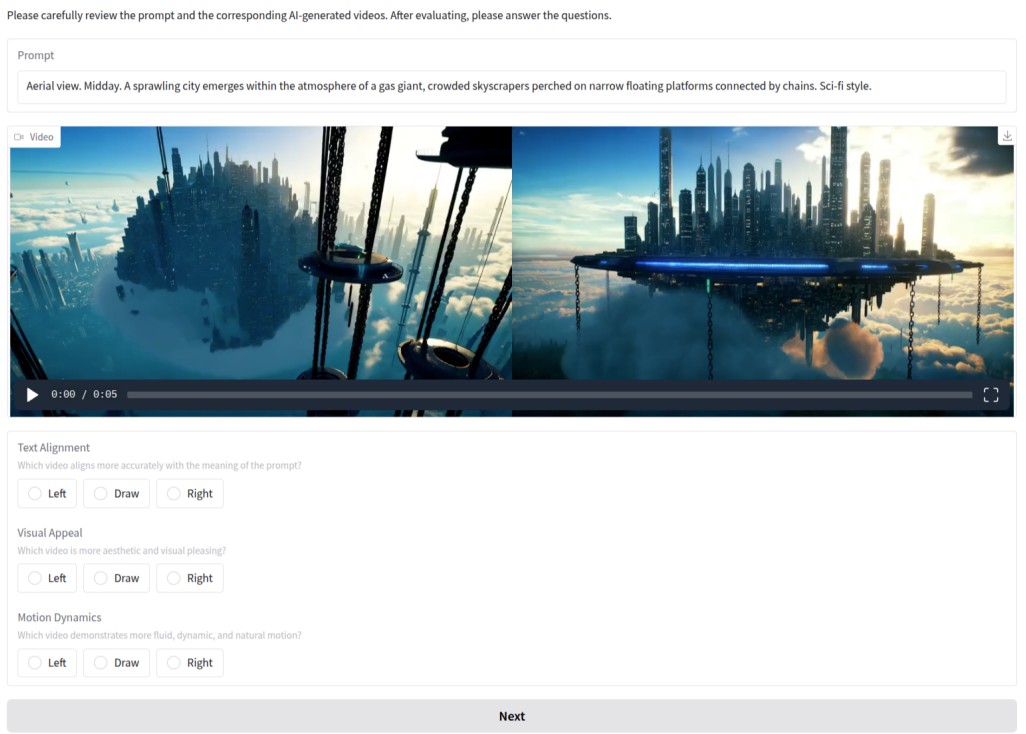

Figure 16: **User preference study interface.** Images and videos are presented in a random order.

# I Additional Qualitative Results

We provide additional qualitative examples in this section. Additional text-to-video results are shown in Figure 17; We present NAG applied to Wan2.1-I2V-14B [6] for image-to-video generation in Figure 18; Figures 19 and 20 demonstrate side-by-side comparisons highlighting fine-grained details; Figures 21 and 22 provide uncurated results to illustrate NAG's general behavior.

**| Wan2.1-T2V-14B, 25 steps**

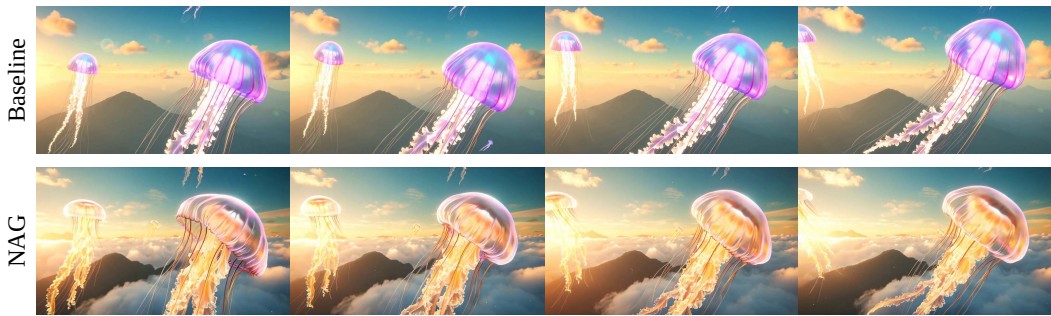

Enormous glowing jellyfish float slowly across a sky filled with soft clouds.
Their tentacles shimmer with iridescent light as they drift above a peaceful mountain landscape.
Magical and dreamlike, captured in a wide shot. Surreal realism style with detailed textures.
– Low resolution, blurry.

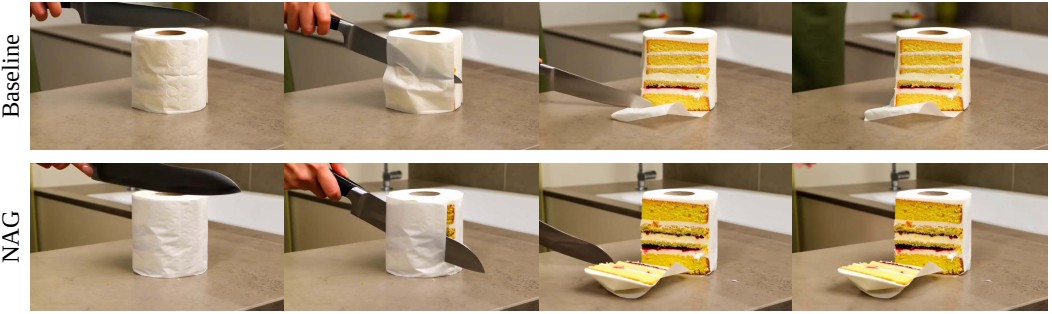

The video begins with a roll of seemingly ordinary toilet paper on a modern bathroom countertop.
As the camera zooms in, a hand holding a knife slices through the roll, revealing it to be a cake.
The cut exposes multiple layers of cake and filling inside. This visual illusion art is both surprising and delightful, blending everyday objects with creative craftsmanship for an engaging and fun reveal.
– Low resolution, blurry.

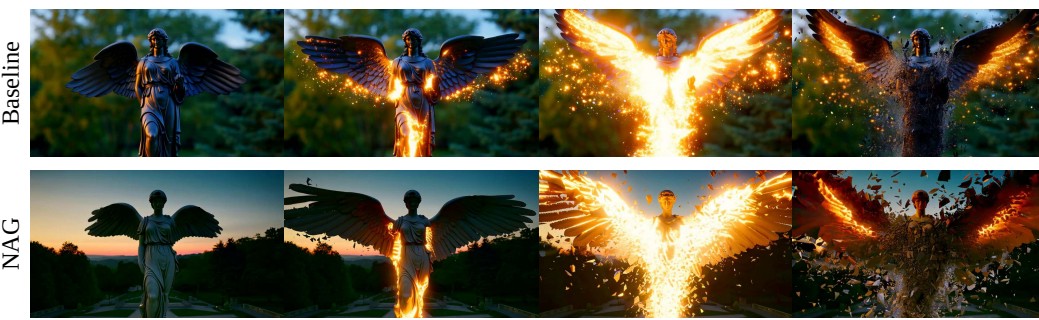

In this video, a statue stands in a picturesque outdoor setting.
At the start, its wings are motionless but gleam under the light.
As the video progresses, the statue begins to transform—cracks appear, glowing effects intensify, and it gradually shatters into countless fragments, creating a breathtaking visual spectacle.
– Static, low resolution, blurry.

Figure 17: **Additional qualitative results of NAG for text-to-video generation.**

**| Wan2.1-I2V-14B, 25 steps**

Reference Image    Prompt

A vibrant red fox with thick fur dashes through a ruin, its bright eyes focused and ears pinned back in full sprint. The illustration, rendered in a fantasy style, uses dynamic blur to simulate a camera tracking the fox's rapid movement, capturing the trail of dust kicked up in its wake. The sense of speed and mystery. This scene is filled with motion and energy.
– Static, low resolution, blurry

Golden sunset bathes a vast snowfield, blending with icy white for a stunning scene. A black sled motorcycle speeds like lightning, its engine roaring through the quiet cold. Snow flies as it leaves tracks behind. The driver, in a heavy black suit, gazes ahead with determination. Distant mountains glow softly under the sun, merging stillness and motion. Close-up captures speed and passion dynamically.
– Static, low resolution, blurry

A tiny ginger cat sits in an open field.  Suddenly, a UFO appears above, emitting a soft, glowing blue beam of light that envelops the cat. the cat is slowly lifted off the ground into the hovering spacecraft. The UFO ascends and disappears, leaving the empty field in silence. Moody, dramatic lighting, atmospheric realism, cinematic visual storytelling.
– Cat, static, low resolution, blurry

Figure 18: **Qualitative results of NAG for image-to-video.**

**| Flux-Dev, 25 steps**

Baseline | NAG

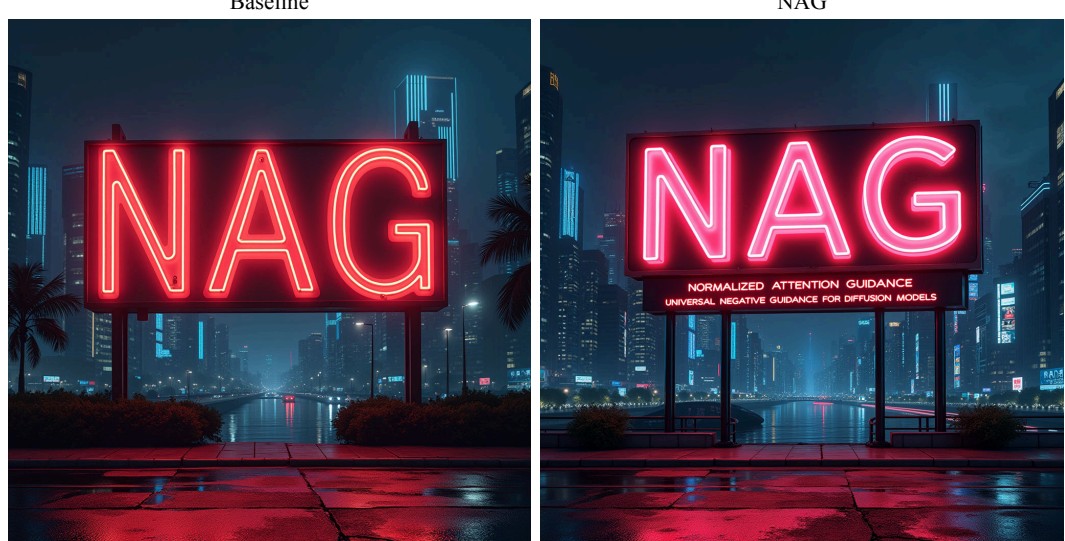

A realistic neon sign at night in a futuristic city. The big letters "NAG"
and the small words "Normalized Attention Guidance: Universal Negative Guidance for Diffusion Models."
- Low resolution, blurry.

**| SD3.5-Large-Turbo, 8 steps**

Baseline | NAG

Close-up portrait of a woman illuminated by soft, warm afternoon light streaming through window blinds,
creating striking shadow patterns across her face. She has a natural, glowing complexion with dewy skin.
The background features muted teal walls that enhance the warmth of the scene. She wears a dark, pinstriped
blazer, adding a touch of elegance. The overall aesthetic is hyperrealistic, capturing intricate details like the
texture of her skin and the delicate play of light and shadow, with a color palette of warm neutrals and soft pastels.
- Low resolution, blurry.

Figure 19: **Detailed qualitative analysis of Flux-Dev [1] and SD3.5-Large-Turbo [15, 52].**

**| Flux-Schnell, 4 steps**

Baseline                                     NAG

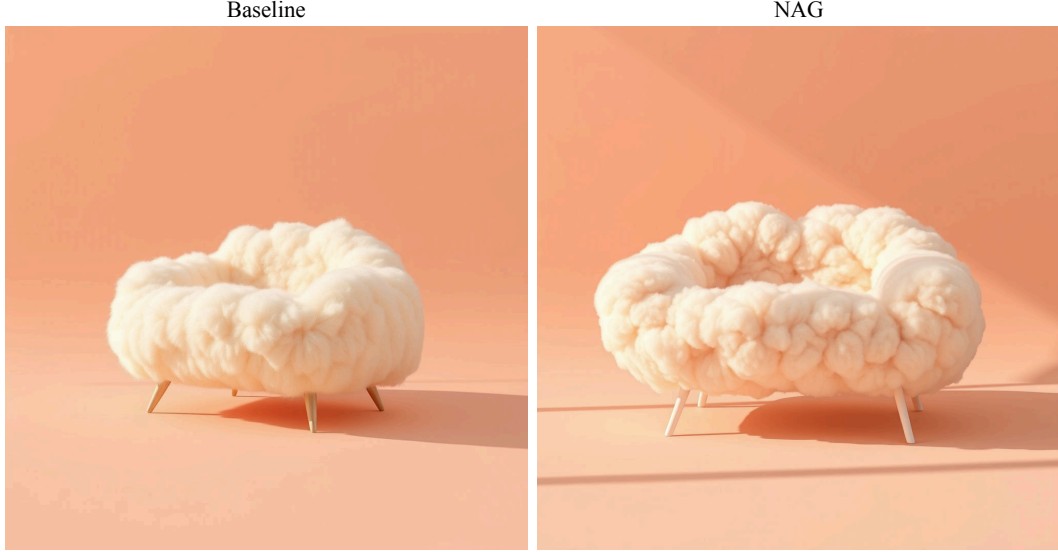

A surreal, modern, small fluffy couch made entirely of pale peach fluffy cloud, shaped like a cumulus cloud.
The chair has four minimalist legs, standing on a seamless solid muted coral background.
Clean and minimalistic, with soft lighting. Ultra high resolution, product design aesthetic, concept art.
Shot by Sony α7R IV, contrast with highlight, casting realism sunlight,
sharp focus, clear detailed, cinematic, glamorous, editorial shoot, 16K, rich detail.
- Low resolution, blurry.

**| SANA-Sprint, 2 steps**

Baseline                                     NAG

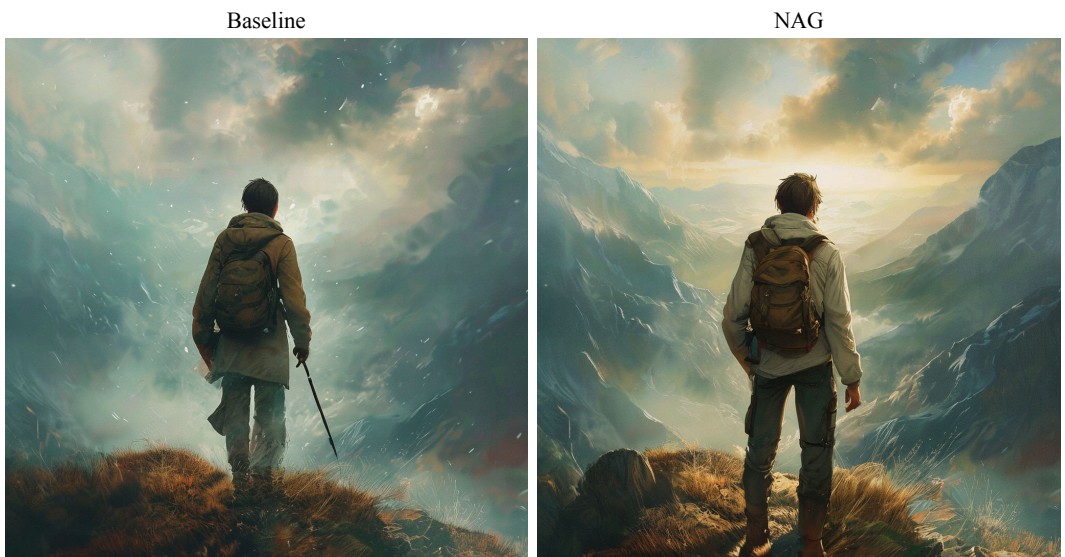

A protagonist, at the beginning of their story, about to embark on their journey.
- Low resolution, blurry.

Figure 20: **Detailed qualitative analysis of Flux-Schnell [1] and SANA-Sprint [53].**

**| Flux-Dev, 25 steps**

| Baseline | NAG | Baseline | NAG | Baseline | NAG | Baseline | NAG |

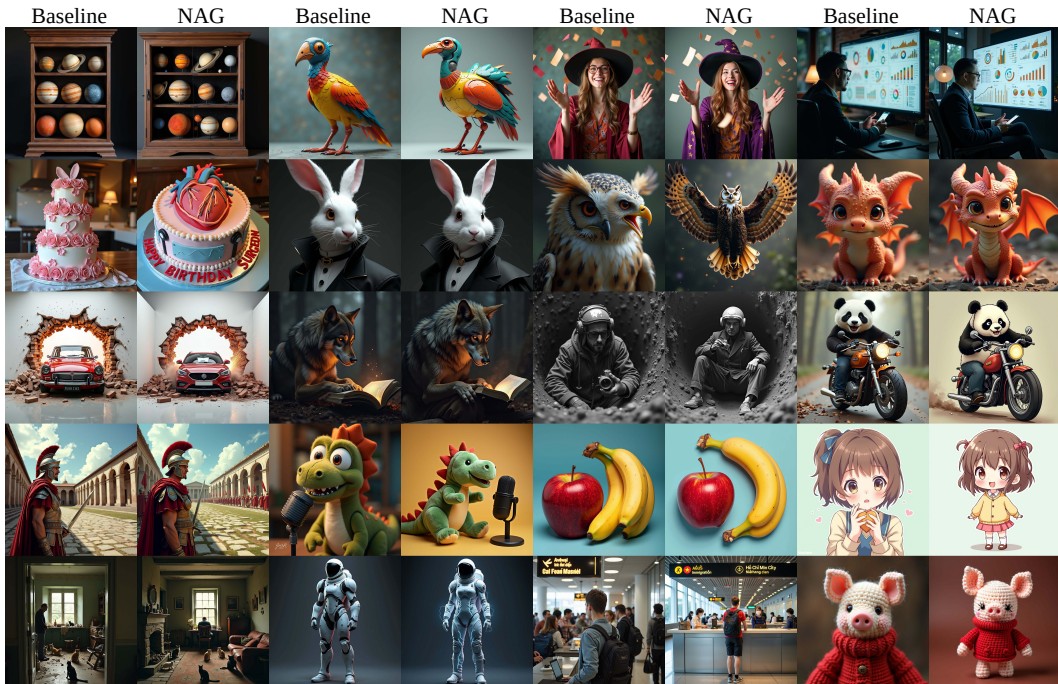

**| SD3.5-Large-Turbo, 8 steps**

| Baseline | NAG | Baseline | NAG | Baseline | NAG | Baseline | NAG |

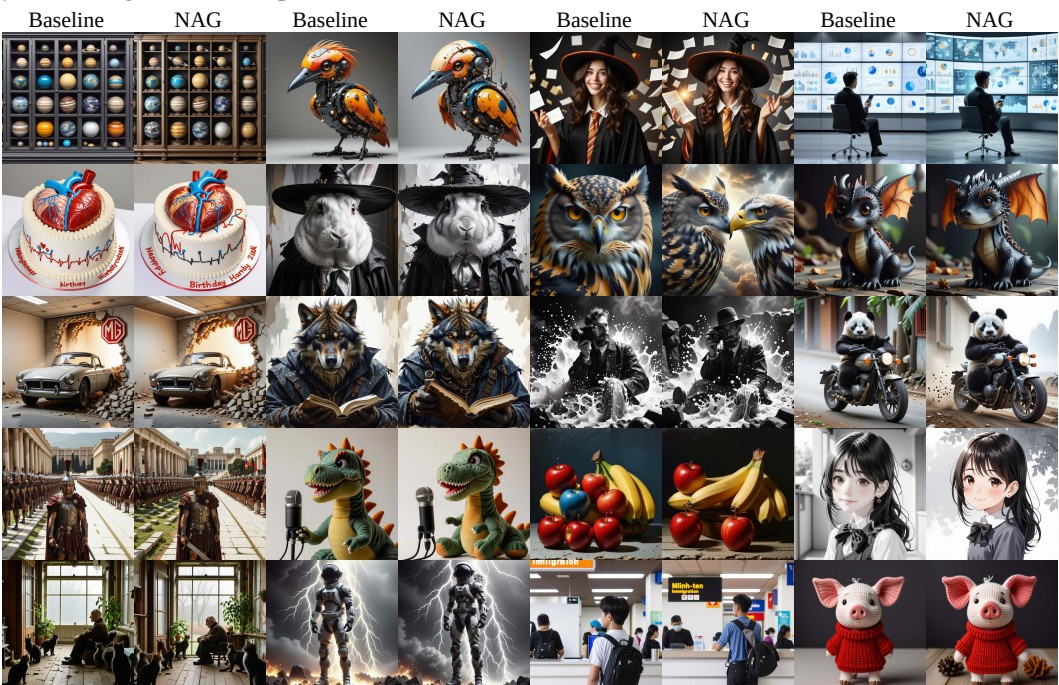

Figure 21: **Uncurated qualitative results for Flux-Dev [1] and SD3.5-Large-Turbo [15, 52].** All prompts are detailed in Appendix J.1. NAG-guided sample utilizes the negative prompt "Low-resolution, blurry."

**| Flux-Schnell, 4 steps**

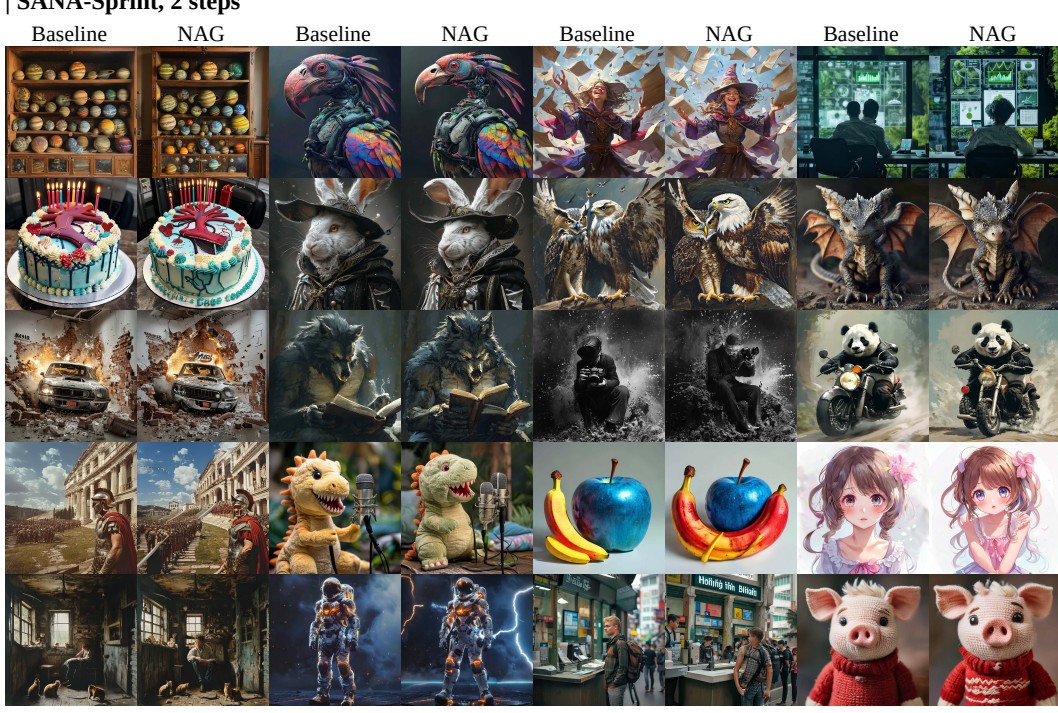

**| SANA-Sprint, 2 steps**

Figure 22: **Uncurated qualitative results for Flux-Schnell [1] and SANA-Sprint [53].** All prompts are detailed in Appendix J.1. NAG-guided sample utilizes the negative prompt "Low-resolution, blurry."

## J Prompt Details

### J.1 Prompts for Figures

**Figure 6**:

1. An anthropomorphic cat thoughtfully paints an oil self-portrait on canvas, capturing its likeness with delicate brushstrokes inside a warmly lit, artistically cluttered studio.
2. An origami fox running in the forest. The fox is made of polygons. Speed and passion. Realistic.

**Figure 21 and Figure 22**:

First row:

1. A cabinet in which all the planets of the solar system are collected
2. a futuristic interpretation of a dodo bird. Cyborg bird. Amazing colorful. Artstation, hyperrealistic
3. a happy female wizard surrounded by pieces of paper flying in the air around her
4. a recruitment consultant, sitting before a screen full of analysis diagram, carrying mobile device, fuji film style, like moss in wandering earth

Second row:

1. Large birthday cake for a cardiothoracic surgeon.
2. an anthropomorphic white rabbit, male wizard face, dressed in black and white, fine art, award-winning, intricate, elegant, sharp focus, cinematic lighting, highly detailed, digital painting, 8 k concept art, art by guweiz and z. w. gu, masterpiece, trending on artstation, 8 k
3. an owl transforms into an eagle
4. a photorealistic dragon pup

Third row:

1. in a room a MGb car smashing through hole in the wall ,sparks dust rubble bricks ,studio lighting,white walls, mg logo
2. a werewolf reading a book
3. Black and white 1905 year portrait of futuristic professional photographer with camera in hand sadly seating deep in a dark pit covered by splash of dust
4. a panda riding a motorcycle

Fourth row:

1. a wide angle photo of roman soldiers in front of courtyard roman buildings,technicolor film ,roman soldier in foreground masculine features nose helmet and silver sword ,eyes,clear sky, arches grass steps field panorama,Canaletto,stone floor,vanishing point,ben-hur flags , a detailed photo, julius Caesar , ceremonial, parade, roma, detailed photo, temples roads hestia,single point perspective, colonnade, necropolis, ultra realism, imperium ,by claude-joseph vernet and thomas cole ,pediment sky clouds,stones in the foreground,dusty volumetric lighting
2. cuddly stuffed dinosaur talking to a microphone
3. blue apple, red banana
4. Anime cute little girl

Fifth row:

1. a lonely man inside a old bucolic house surrounded by cats by Richard Billingham
2. full body space suit with boots, futuristic, character design, cinematic lightning, epic fantasy, hyper realistic, detail 8k

3. a young danish traveller standing at an immigration counter in ho chi minh city
4. Amigurumi figure of a little pig wearing a red sweater, professional photography, close up, vintage, 8k, product photo

## J.2 Prompts for User Study

Table 10: **Positive-negative prompt pairs used in the user study comparing NAG and NASA.**

| Positive prompt | Negative prompt |
|---|---|
| A photo of a person | Male 
 Female |
| A photo of a person | Young 
 Old |
| A photo of a person | Dark 
 Bright |
| A photo of a pet | Cat 
 Dog |
| A photo from the 1960s | Black and white 
 Color |
| A still painting of fruits | Apples 
 Grapes |
| A landscape | Trees 
 Mountains |
| A room with furniture | Bed 
 Window |

