# OpenReview forum: "Normalized Attention Guidance: Universal Negative Guidance for Diffusion Models"
_NeurIPS.cc/2025/Conference — NeurIPS 2025 poster_

### Official Review · Reviewer_zEeN · 2025-06-16

**Clarity:** 2
**Significance:** 2
**Originality:** 2
**Rating:** 4
**Confidence:** 2

**Summary:**

This paper introduces Normalized Attention Guidance (NAG), a novel method for negative guidance in diffusion models. It addresses the limitations of existing approaches, such as Classifier-Free Guidance (CFG), which fail in few-step sampling regimes. NAG operates directly within the attention feature space, applying controlled extrapolation complemented by L1-norm-based normalization and feature refinement to prevent feature drift issues. Extensive experiments demonstrate its robustness and ability to generalize effectively.

**Questions:**

None

**Ethical Concerns:**

["NO or VERY MINOR ethics concerns only"]

**Final Justification:**

The rebuttal has satisfactorily addressed my concerns, and I lean toward accepting the paper.

**Limitations:**

- While NAG operates effectively in attention feature space, its applicability may be limited to models or architectures that leverage attention mechanisms. It may not generalize to non-attention-based architectures or other types of generative models.

**Quality:**

2

**Strengths And Weaknesses:**

Strengths
- This paper is well written and easy to follow.
- The authors conduct comprehensive experiments across a wide range of models (UNet, DiT), sampling regimes (few-step to 25-step), and modalities (image and video). Metrics used include CLIP Score, FID, PFID, and ImageReward, providing multi-faceted validation.
- NAG adds minimal computational overhead and does not require retraining, making it highly practical for deployment.

Weaknesses
- As shown in Figure 8, the effect of normalization does not appear to be significant. Additionally, the experiment "w/o Norm & with Refine" is missing.
- While generally effective, the paper acknowledges that NAG struggles with very aggressive prompts or stylistic suppression, which may lead to instability or visual artifacts (Figure 11).

---

> ### Author Rebuttal · Authors · 2025-07-31
>
> We thank for acknowledging the paper’s clear presentation and comprehensive evaluation of NAG. Below we address the two points with new ablations and expanded applicability experiments.
>
> ### As shown in Figure 8, the effect of normalization does not appear to be significant. Additionally, the experiment "w/o Norm & with Refine" is missing.
>
> Thank you for catching the missing ablation. We have now added the “w/o Norm” ablative study alongside the Full NAG:
>
> **$\phi = 4$**
>
> | Flux-Schnell 4-step | CLIP ↑ | FID ↓ | PFID ↓ | ImageReward ↑ |
> |:--------------------:|:---:|:---:|:----:|:-----------:|
> | Full NAG            |   32.0  |   24.46  |   34.95   |      1.099       |
> | w/o Norm.           |   31.9  |   25.78  |   37.32   |      1.107       |
>
> **$\phi = 8$**
>
> | Flux-Schnell 4-step | CLIP ↑ | FID ↓ | PFID ↓ | ImageReward ↑ |
> |:--------------------:|:---:|:---:|:----:|:-----------:|
> | Full NAG            |   32.2  |   24.13  |   33.92   |      1.069       |
> | w/o Norm.           |   32.1  |  26.12   |   37.22   |      1.081       |
>
> Removing normalization degrades both fidelity metrics (FID, PFID), and leads to even larger drops at higher scales. The degraded fidelity indicates that the guidance becomes unstable and disrupts the feature flow through the network. Normalization therefore acts as a critical safeguard by bounding feature magnitudes.
>
> ### While NAG operates effectively in attention feature space, its applicability may be limited to models or architectures that leverage attention mechanisms.
>
> Actually NAG is not inherently tied to attention mechanism. Because it exploits the semantic difference between two conditions, the same guidance algorithm can be applied to any conditioning module.
>
> We examine two conditioning mechanisms in SDXL: cross‑attention and adaptive normalization.
>
> - Adaptive‑Norm Only: applying NAG just in adaptive normalization layer already yields consistent gains.
> - Attention + Adaptive‑Norm: There is no obvious improvement compared to attention only NAG, implying that guidance in adaptive normalization contributes minimal additional benefit over attention space NAG.
>
> We focused our paper on attention layers because it is the dominant conditioning mechanism in today’s generative models, but these results clearly demonstrate that NAG can be ported to different conditioning mechanisms without modification of the core algorithm.
>
> | SDXL-DMD2 4-step | CLIP ↑ | FID ↓ | PFID ↓ | ImageReward ↑ |
> |:-----------------:|:---:|:---:|:----:|:-----------:|
> | Baseline         |  31.6   |  24.79   |   27.11   |         0.876    |
> | NAG (adaptive)   |  31.8   |  24.73   |   26.79   |         0.892    |
> | NAG (attention)  |  **32.2**   |  23.32   |   **25.61**   |         **0.960**    |
> | NAG (attention + adaptive)     |  **32.2**   |   **23.27**  |   25.62   |    0.945         |

---

> > ### Comment · Reviewer_zEeN · 2025-08-07
> >
> > Thanks for the detailed response to my comments. I am satisfied with the additional ablation without normalization and with refinement, the effect of normalization on the fidelity metrics, and the potential applicability to different conditioning mechanisms. My concerns have been adequately addressed.

---

### Official Review · Reviewer_63rU · 2025-06-28

**Clarity:** 3
**Significance:** 2
**Originality:** 1
**Rating:** 3
**Confidence:** 4

**Summary:**

This paper introduces Normalized Attention Guidance (NAG), a plug-in technique for attention-based diffusion models that achieves effects similar to negative prompting through simple attention modulation. The method employs L1 normalization and linear combination to process attention maps from both unconditional and conditional branches.

**Questions:**

- Why does modulating self-attention outputs preserve generation quality and not corrupt the learned data manifold?
- Why specifically L1 normalization? Do you have any ablation studies comparing other normalization variants? There are also variants that have as benign an effect as L1 normalization on low magnitude signals.
- If NAG still requires CFG, what fundamental classes of problems does it solve that CFG alone cannot? If there's no such distinction, why not study the prediction of classifier-free guidance and then modulate attention layers within it?

**Ethical Concerns:**

["NO or VERY MINOR ethics concerns only"]

**Final Justification:**

After reading other reviewers' comments and the authors' responses, several of my concerns have been resolved. However, concerns about the incremental contribution remain. I have decided on a final rating of borderline reject but do not strongly oppose acceptance.

**Limitations:**

yes

**Quality:**

2

**Strengths And Weaknesses:**

## Strengths
- This paper provides a straightforward approach through normalization and linear combination of attention maps.
- Since the method is based on simpler assumptions, this paper shows broad applicability across diffusion model variants, including single-step and video generation tasks.
- NAG can handle edge cases where CFG with negative prompts cannot.

## Weaknesses
- Incremental contribution is of major concern. The core innovation appears to be adding a normalization step to NASA, which represents an incremental rather than fundamental contribution.
- Modulating self-attention outputs fundamentally alters the learned data manifold, yet the paper lacks rigorous analysis of why these modifications preserve generation quality. Figure 4 somewhat addresses this, and I understand that L1 normalization constraints may limit extreme deviations, but this doesn't fully justify why such modulation is admissible.
- Unlike Classifier-Free Guidance (CFG), which derives from well-established Classifier Guidance principles enabling principled extrapolation of noise predictions, this approach lacks comparable theoretical grounding.
- The technique still requires CFG integration rather than serving as a standalone replacement, raising questions about its fundamental necessity.
- Ablation studies comparing L2 normalization and max normalization variants are absent, limiting understanding of the method's sensitivity to normalization choices.

---

> ### Author Rebuttal · Authors · 2025-07-31
>
> Thank you for recognizing NAG’s broad applicability and for these insightful questions.
>
> ### Q1: Why does modulating self-attention outputs preserve generation quality and not corrupt the learned data manifold?
>
> To clarify, NAG operates on cross-attention or multimodal DiT (MMDiT) layers, not on self-attention layers; the notation $Z$ in our paper denotes the cross-attention features (Equation 4).
>
> As detailed in our response to Reviewer oLD6, NAG can be viewed as a variant of CFG’s implicit classifier gradient applied at intermediate cross-attention layers rather than only at the final output. NAG performs a series of small, implicit classifier‑guided steps where the positive and negative features remain semantically aligned, preventing runaway divergence from the manifold. The subsequent normalization and refinement bounds feature magnitudes near the model’s learned feature space, alleviating the risk of manifold corruption.
>
> Moreover, these modifications are continuously smoothed by the model’s linear projections, LayerNorm, nonlinear activations, and other layers before the next attention operation. These architectural “filters” mix, re-center, and softly cap value spikes, preventing outlier attention features from cascading into large distributional shifts. distributional shifts.
>
> ### Q2: Why specifically L1 normalization? Do you have any ablation studies comparing other normalization variants?
>
> Thank you for pointing this out! We have run ablations comparing L1 against L2 and max-norm, but did not include them in the submission; we will add these results and accompanying discussion in the paper.
>
> We find that swapping to L2 yields visually similar results but with slightly worse metrics, while max‑norm will soften fine details.
> Overall, any norm scales the attention features, providing constraints without changing semantic direction, so NAG’s performance is not really sensitive to the exact choice.
>
> L1 was chosen because it conceptually best preserves proportional relationships and subtle semantics (lines 167 - 170), provides the best quantitative results, and it’s the most efficient to compute.
>
> | Flux-Schnell 4-step | CLIP ↑ | FID ↓ | PFID ↓ | ImageReward ↑ |
> |:--------------------:|:---:|:---:|:----:|:-----------:|
> | NAG (L1)            |   **32.0**  |   **24.46**  |   **34.95**   |      1.099       |
> | NAG (L2)            |   31.9  |   25.20 |   36.93   |       **1.104**      |
> | NAG (Max)           |   31.9  |   25.35  |   37.18   |      1.089       |
>
> ### Q3: If NAG still requires CFG, what fundamental classes of problems does it solve that CFG alone cannot? If there's no such distinction, why not study the prediction of classifier-free guidance and then modulate attention layers within it?
>
> We apologize for the lack of clarity. NAG works completely independently under few‑step sampling (Figure 6, Table 5), where CFG collapses (Left of Figure 3). In essence, NAG serves as a standalone alternative to CFG for negative guidance in few-step sampling.
>
> Under multi-step sampling, NAG can also be used in conjunction with CFG. When paired with CFG, NAG further enhances qualitative and quantitative results (Figure 7, Table 6). This dual capability highlights NAG’s universality.

---

> > ### Comment · Reviewer_63rU · 2025-08-05
> >
> > Thank you to the authors for the clarification. The response addressed several of my concerns. I remain somewhat concerned about the incremental contribution beyond NASA, but I would not strongly oppose acceptance.

---

### Official Review · Reviewer_P6ev · 2025-07-02

**Clarity:** 3
**Significance:** 2
**Originality:** 2
**Rating:** 4
**Confidence:** 4

**Summary:**

This paper builds on NASA [1], which is an attention guidance/extrapolation method for negative prompting, by applying two additional operations: (1) feature normalization by clipping and re-scaling the extrapolated feature, and (2) feature blending via interpolating between the re-scaled feature and the positive guidance feature. These constraints restrict the features to be within the manifold of meaningful representations, while maintaining the effectiveness of negative guidance. It is a training-free, simple, and effective method. Experiments are conducted on different architectures (UNet and DiT), different sampling steps, and on both image and video generations, demonstrating the adaptability of the proposed method.

[1] SNOOPI: Supercharged One-step Diffusion Distillation with Proper Guidance. Arxiv 2024.

**Questions:**

I am happy to raise the score if the authors can sufficiently address the following questions and the concerns in the weaknesses section:

1. Any performance gaps when the guidance scale $\phi<1$ (i.e., doing interpolation)? From Figure 2, it seems the gap only exists when the scale is large, while one can still have effective results by using the baseline NASA method by simply setting this hyperparameter properly.
2. Have you experimented with the proposed method on stronger few-step generation DiT and UNet backbones to show consistent performance enhancement to support the claim that NAG can be uniformly applied?
3. Can you provide analyses of the hyperparameters additionally introduced by NAG?
4. Can you provide additional analysis of the proposed two stabilization mechanisms to demonstrate the technical contribution in this paper in addition to NASA [1]?

**Ethical Concerns:**

["NO or VERY MINOR ethics concerns only"]

**Final Justification:**

The rebuttal has well addressed my listed concerns. I am now more inclined to recommend acceptance of this paper.

**Limitations:**

yes

**Quality:**

3

**Strengths And Weaknesses:**

Strengths

1. The writing of the paper is clear and well-structured, making the paper easy to follow and understand. The visual examples are helpful in illustrating the limitation of the standard CFG and the baseline NASA method.
2. The proposed NAG is training-free, simple and effective according to several metrics such as CLIP Score, FID, PFID, and user study.
3. The proposed NAG is adaptable to different settings of diffusion models, such as different architectures (UNet and DiT), different sampling steps (one-step and few-steps), and different modality (image and video generation).

Weaknesses

1. The improvements over the NASA [1] baseline is somewhat incremental, as only the last two operations (normalization and blending) are additionally proposed in NAG compared to NASA. Also, these two operations are more like tricks instead of technical improvements. Specifically, the normalization step is essentially doing a clipping/rescaling, while the blending step (Equation 10) seems to up-weight the positive branch from Equation 7, whose effect may be achieved by additional hyperparameter tuning in Equation 7. In such case, I would suggest the authors conduct more analysis (include but not limited to why L1 can retain essential structure for precise guidance as claimed) to show the technical contribution of these two operations, which should be emphasised more than the design of attention-based guidance by applying extrapolation in attention feature space, which NASA uses. Currently, such analysis is quite insufficient, as shown in the short methodology section 4.1.
2. The proposed method introduces an additional blending scale hyperparameter $\alpha$. It would be good to see additional analysis of the performance sensitivity to this when it is set to 0.1, 0.3, … 0.9, which is currently missing. Also, during the normalization step, an additional clipping threshold $τ$ is employed to limit feature magnitude. The sensitivity analysis of such hyperparameter is also missing.
3. In Figure 1, the caption includes “CFG fails in few-step models”, which is, however, not reflected in the Figure. It’d be better if the authors could include the failure examples of CFG for these specific examples for qualitative comparison, or simply remove the caption to reduce the emphasis on comparing with CFG and make it a standalone plot for demonstrating the result of the proposed NAG.
4. The denominator R in Equation 9 is not defined. Should it be the previously defined R[i]?

---

> ### Author Rebuttal · Authors · 2025-07-31
>
> We appreciate the suggestion to illustrate CFG failures in Figure 1. And we will correct the notation around $R[i]$ to ensure clarity.
>
>
> ### Q1: Any performance gaps when the guidance scale $\phi < 1$ (i.e., doing interpolation)? From Figure 2, it seems the gap only exists when the scale is large, while one can still have effective results by using the baseline NASA method by simply setting this hyperparameter properly.
>
> For the UNet SDXL‑DMD2 model, we include an extended comparison against NASA in Appendix F (lines 530–541), covering both quantitative metrics and a user study, which illustrate improvements of NAG over NASA across all evaluated criteria.
>
> Additionally, we evaluate DiT Flux‑Schnell under small guidance scales here. Importantly, $0<\phi<1$ does not strictly correspond to interpolation—it simply applies a lighter guidance strength. In the experiments, NASA at low $\phi$ does improve CLIP score but degrades FID and ImageReward, and crash in $\phi = 0.5$. In contrast, NAG demonstrates stable enhancements and achieves the best overall metrics on $\phi = 4$, boosting both alignment and image quality. As a result, NAG is robust to a wide range of $\phi$, significantly reducing tuning overhead.
>
> Metrics reported for Baseline / NAG.
>
> | Flux-Schnell 4-step | CLIP ↑ | FID ↓ | PFID ↓ | ImageReward ↑ |
> |:--------------------:|:---:|:---:|:----:|:-----------:|
> | Baseline        |  31.4  |  25.47   |   38.26   |      1.029       |
> | $\phi$=0.125        |  31.7 / 31.5  |  27.10 / 25.48   |   36.49 / 38.15   |      0.929 / 1.035       |
> | $\phi$=0.25         |  31.7 / 31.5   |  33.04 / 25.53   |   36.65 / 38.11   |      0.591 / 1.042       |
> | $\phi$=0.5          |  21.7 / 31.5   |  297.36 / 25.50   |   305.3 / 37.75   |      -2.08 / 1.057       |
> | $\phi$=4            |  - / **32.0**   |  - / **24.46**   |   - / **34.95**   |      - / **1.099**       |
>
> ### Q2: Have you experimented with the proposed method on stronger few-step generation DiT and UNet backbones to show consistent performance enhancement to support the claim that NAG can be uniformly applied?
>
> Beyond the experiments in Section 5.1 of the main paper, we have evaluated NAG on a wide range of few‑step backbones in Appendix D (lines 524–529). Overall, we include SOTA models from academia and the community, such as DiT architecture Flux, SD3.5-Turbo (SIGGRAPH Asia 2024) and SANA-Sprint (ICCV 2025), as well as UNet architecture NitroFusion (CVPR 2025), DMD2 (NeurIPS 2024) and HyperSD (NeurIPS 2024). On average, we can observe +0.5 CLIP, −1.5 FID, -2.5 PFID, and +0.1 ImageReward gains.
>
> Additionally, in the rebuttal to the Reviewer oLD6, we demonstrate improved performance, -1.3 FID and +20 IS, for the newly released single-step class‑conditional MeanFlow.
>
> ### Q3: Can you provide analyses of the hyperparameters additionally introduced by NAG?
>
> We include an ablation study on hyperparameters in Appendix F (lines 542–552). In particular, Figure 13 visually analyzes the performance sensitivity of normalization threshold $\tau$ and refinement factor $\alpha$. We hope this analysis, along with visual results, clarifies NAG’s sensitivity to hyperparameter choices.
>
> ### Q4: Can you provide additional analysis of the proposed two stabilization mechanisms to demonstrate the technical contribution in this paper in addition to NASA?
>
> Unlike NASA, which directly subtracts negative attention maps, NAG applies extrapolation grounded in the implicit classifier view (please refer the Q1 of reviewer oLD6 for more details). This yields more stable editing in attention space. The subsequent normalization and refinement further reinforce this stability.
>
> These operations share two critical properties:
>
> 1. Firm Nonexpansiveness: Each update can only bring features closer together, never amplifying deviations beyond the model’s learned manifold.
> 2. Direction Preservation: By projecting onto norm‑balls or taking convex combinations, the original edit direction remains intact, ensuring semantic coherence.
>
> While these properties guarantee features stay near the learned manifold, they do not ensure that NAG's attention space edits will always yield a stable guidance signal.
>
> Empirically, however, our ablations confirm that both normalization and refinement are indispensable for stability. In practice, NAG remains a simple yet powerful mechanism for robust and controllable negative guidance across architectures and sampling regimes.
>
> We believe NAG is just the beginning, and encourage future research to explore alternative attention space constraint formulations to achieve more stable guidance.

---

> > ### Comment · Reviewer_P6ev · 2025-08-05
> >
> > I appreciate the authors' rebuttal, which has well-addressed my concerns, and I am happy to raise my rating accordingly.

---

### Official Review · Reviewer_oLD6 · 2025-07-06

**Clarity:** 3
**Significance:** 3
**Originality:** 2
**Rating:** 4
**Confidence:** 5

**Summary:**

This paper proposes Normalized Attention Guidance (NAG), a novel inference-time approach for negative prompting in diffusion models. Unlike traditional Classifier-Free Guidance (CFG) or Perturbed Attention Guidance (PAG) methods that operate in the output space, NAG directly performs extrapolation within the attention feature space between positive and negative conditioned attention maps. To ensure stable extrapolation, the authors introduce L1-based normalization and refinement blending, constraining attention vectors within stable manifolds. Comprehensive experiments demonstrate the effectiveness and generalizability of NAG across multiple model architectures (UNet, DiT), sampling regimes (few-step, multi-step, guidance distilled models), and modalities (image, video).

**Questions:**

Q1. Beyond geometric interpretation, can authors provide a formal theoretical justification or probabilistic interpretation for extrapolating directly in attention feature space? Is there any formal rationale for why this manipulation maintains semantic coherence and stability?

Q2.Considering PLADIS (ICCV 2025) already introduced attention-space extrapolation, how precisely does NAG differ methodologically from PLADIS? Is NAG essentially a special case of PLADIS, with sparse positive and dense negative attention maps, or are there fundamental differences beyond the application task?

Q3. What specific hyperparameter values (particularly the refinement blending factor α) are used across experiments, and why were these particular choices made? How sensitive is performance to variations in α?

Q4. Given the empirical observations in Figures 8 and 9 showing peak performance at relatively low guidance scales (around 2–4), what is the practical advantage of employing the additional complexity of normalization and refinement mechanisms to enable higher guidance scales?

Q5. How does NAG perform independently without CFG or PAG? Can standalone NAG achieve competitive results, or is the method inherently complementary rather than independently effective?

Q6. While NAG is designed explicitly for negative prompting, can the authors extend or generalize the NAG approach to standard (positive-only) text-to-image generation scenarios, without explicitly provided negative prompts?

**Ethical Concerns:**

["NO or VERY MINOR ethics concerns only"]

**Final Justification:**

During the rebuttal period, the author provided both experimental and theoretical findings to address my remaining concerns. Therefore, I am voting for a positive rating.

**Limitations:**

- Theoretical underpinnings beyond geometric intuition are completely absent, weakening interpretability.

- Novelty claims must be revised considering PLADIS’s prior introduction of attention-space extrapolation.

- Lack of standalone evaluations limits the clarity of intrinsic method effectiveness.

- Incomplete quantitative assessments (particularly video evaluation) and specialized metrics are significant gaps.

**Quality:**

4

**Strengths And Weaknesses:**

\** Strength

\#1 Attention-space Extrapolation:
NAG’s core innovation, extrapolating directly in attention feature space, offers rich semantic control over diffusion models, surpassing output-space extrapolation approaches in terms of stability and flexibility.

\#2 Strong Empirical Generalization:
Extensive experiments show NAG's generalizability across architectures, sampling scenarios (including aggressive few-step regimes and distilled models), and modalities such as images and videos, which are clearly valuable contributions.

\#3 Practicality and Ease of Integration:
NAG is computationally lightweight, training-free, and easily integrated into existing diffusion pipelines. These practical benefits significantly enhance the approach's applicability.

\** Major Weakness

\#1 Lack of Theoretical Grounding beyond Geometric Interpretation

- While the authors provide a helpful geometric interpretation of attention-space extrapolation, they overly rely on this intuition without offering formal theoretical analysis or probabilistic justification.

- CFG and PAG clearly interpret extrapolation in output space within a well-understood diffusion probabilistic framework. In contrast, extrapolation in attention space fundamentally differs and lacks an equivalent probabilistic interpretation.

The authors exclusively rely on empirical observations and geometric explanations without theoretical guarantees or detailed formal analysis, making the underlying principles unclear.

\#2  Inaccurate Novelty Claim regarding Attention-Space Extrapolation
The authors claim NAG is the first method performing extrapolation in attention feature space. However, the method presented in:
- PLADIS: Pushing the Limits of Attention in Diffusion Models at Inference Time by Leveraging Sparsity (ICCV 2025)
already introduced attention-space extrapolation explicitly. Thus, novelty claims must be significantly narrowed or revised clearly.

- Indeed, NAG seems conceptually a special case of PLADIS, where positive attention maps are sparse and negative attention maps are dense. Explicit citation of PLADIS and detailed methodological differences are crucial, not just differences in tasks or applications.

\#3 Insufficient Justification of Normalization and Refinement Components
While the normalization and blending (refinement) steps are central to NAG’s success, explicit hyperparameter details—particularly the alpha (α) value for refinement blending—are not clearly reported in the main text.

- The manuscript fails to explain intuitively or theoretically why normalization and refinement specifically enhance stability.

- Although the authors argue these mechanisms allow higher guidance scales, results in Figure 8 show performance peaks around relatively modest scales (2–4), questioning the practical value of further increasing the guidance scale (e.g., to 16). Qualitatively, Figure 9 similarly indicates diminishing returns after moderate scales, undermining this core claimed advantage.

\#4 No Standalone Performance Evaluation for NAG
All provided experimental results combine NAG with other guidance methods (CFG or PAG).

- Critical information on NAG's standalone performance without additional guidance methods is absent, limiting understanding of NAG’s intrinsic value and effectiveness.

\** Minor Weakness

\#1  Lack of Specialized Quantitative Metrics:
The current evaluation relies heavily on general-purpose metrics like ImageReward. More specialized metrics specifically designed for diffusion and text-image alignment (e.g., PickScore, HPSv2) would provide deeper insights into alignment quality.

\#2 Lack of Quantitative Video Evaluation:
Video generation results rely solely on qualitative assessments, lacking quantitative metrics such as VBench. Providing quantitative results would strengthen reliability and interpretability.

---

> ### Author Rebuttal · Authors · 2025-07-31
>
> We’re grateful for the thorough reviews and recognition of our strong empirical generalization.
>
> ### Q1: Beyond geometric interpretation, can authors provide a formal theoretical justification or probabilistic interpretation for extrapolating directly in attention feature space? Is there any formal rationale for why this manipulation maintains semantic coherence and stability?
>
> Below, we provide a theoretical interpretation for NAG attention space extrapolation and will add it to the paper.
>
> **Implicit Classifier Gradient**
>
> For a pretrained diffusion network $G_\theta$, and timestep $t$ we denote the conditional and unconditional noise predictions as
>
> $\hat\epsilon = G_\theta(x_t, c),\qquad\hat\epsilon^\varnothing =G_\theta(x_t, \varnothing).$
>
> By Bayes’ rule, the implicit classifier over $x_t$ can be defined as
>
> $p^i(c\mid x_t) \propto \frac{p(x_t\mid c)}{p(x_t\mid \varnothing)}.$
>
> Its gradient can be approximated with
>
> $\nabla_{x_t}\log p^i(c\mid x_t)= \nabla_{x_t}\log p(x_t\mid c) - \nabla_{x_t}\log p(x_t\mid \varnothing) \approx -\bigl(\hat\epsilon - \hat\epsilon^{\varnothing}\bigr).$
>
> CFG then performs a heuristic gradient ascent step, leading to the guidance scaled by $\phi$
>
> $\tilde\epsilon= \hat\epsilon + \phi \bigl(\hat\epsilon - \hat\epsilon^\varnothing\bigr).$
>
> **Generalizing to Intermediate Features**
>
> Rather than only adjusting the final noise estimate, we can view each intermediate feature map $F_l$ of layer $l$ as also carrying implicit classifier gradient. Denote
>
> $F_l = G_\theta^l(x_t, c),\qquad F_l^{\varnothing} = G_\theta^l(x_t, \varnothing).$
>
> Analogously, we approximate
>
> $\nabla_{x_t}\log p^i(c\mid x_t)\approx -\bigl(F_l - F_l^{\varnothing}\bigr),$
>
> so an ascent step in feature space yields
>
> $\widetilde F_l= F_l + \phi \bigl(F_l - F_l^{\varnothing}\bigr).$
>
> **Negative Guidance**
>
> To steer generation *away* from unwanted attributes, we replace the null condition with negative condition $c^-$. Given
>
> $F_l^+ = G_\theta^l(x_t, c^+),\qquad F_l^- = G_\theta^l(x_t, c^-),$
>
> the unified negative guidance becomes
>
> $\widetilde F_l = F_l^+ + \phi \bigl(F_l^+ - F_l^-\bigr).$
>
> This directs $F_l$ both toward the positive semantics and away from negative features in one update.
>
> **Layer‑Wise Updates for Stability**
>
> CFG and NAG can be viewed as two special cases of where to apply the unified guidance.
>
> - CFG (output‑space) applies a single global adjustment at the final layer. Under aggressive few‑step sampling, the positive and negative predictions have already drifted apart, causing severe artifacts.
> - NAG (attention‑space) instead interleaves guidance steps in attention layers. Because $F_l^+$ and $F_l^-$ remain closer in feature space early on, each update is well‑conditioned and prevents runaway divergence.
> The cumulative effect of many modest, layer‑wise adjustments retains semantic coherence and provides stable negative guidance even with very few denoising steps.
>
> By viewing each attention map as contributing an implicit gradient, NAG generalizes CFG’s single output guidance into a sequence of feature space guidance.
>
>
> ### Q2: Considering PLADIS (ICCV 2025) already introduced attention-space extrapolation, how precisely does NAG differ methodologically from PLADIS?
>
> We appreciate the pointer to PLADIS (published in March 2025) and we will include it and clarify the distinctions in our related work.
>
> While PLADIS also operates in the cross‑attention domain, it contrasts dense versus sparse attention maps extracted from a single prompt. PLADIS treats the dense map as a perturbation of its sparse counterpart, in a manner reminiscent of Perturbed‑Attention Guidance (PAG).
>
> By contrast, NAG explicitly exploits the semantic difference between a positive prompt and a negative prompt, providing targeted suppression of undesired attributes during generation, a capacity that PLADIS and PAG do not support. When a negative prompt is absent, NAG reduces to the perturbation‑style mechanism by treating the null condition as the denser attention reference. This dual capability sits naturally within our unified negative‑guidance framework.
>
>
>
> ### Q3: What specific hyperparameter values (particularly the refinement blending factor α) are used across experiments, and why were these particular choices made? How sensitive is performance to variations in α?
>
> All default NAG hyperparameters, including the refinement blending factor $\alpha$ and normalization threshold $\tau$, are listed in Appendix C (lines 521–523). Defaults $\tau = 2.5$, $\alpha \in [0.125, 0.5]$ vary by model family. In Appendix F (lines 542–552), we present an ablation study over $\alpha$ and $\tau$, demonstrating that performance remains stable for values of $\alpha$ below $0.5$ and $\tau$ around $2.5$.
>
> Although a full grid search across every model family would be computationally prohibitive, we conducted smaller‑scale explorations to identify an effective range, then chose the defaults based on visual effect. In practice, users may begin with these settings and adjust them as needed to suit personal preferences.
>
> ### Q4: What is the practical advantage of employing the additional complexity of normalization and refinement mechanisms to enable higher guidance scales?
>
> High guidance scales often introduce visual artifacts or even collapse, limiting how aggressively users can steer generation. NAG’s normalization and refinement steps prevent these artifacts. As a result, users can apply much more aggressive negative guidance without risk, achieving robust suppression of unwanted attributes.
>
> ### Q5: How does NAG perform independently without CFG or PAG?
>
> NAG was designed to work as a standalone negative guidance in few‑step (1–8 steps) sampling where CFG and PAG fail (lines 54-55).
>
> In our few‑step experiments (Section 5.1) on both UNet and DiT backbones, applying NAG by itself (without CFG or PAG) demonstrates substantial gains in CLIP alignment, FID/PFID, and ImageReward compared to the baseline.
> In multi‑step regimes (Section 5.3), NAG can be combined with CFG or PAG for even stronger guidance, but on few‑step models NAG is the only method that delivers universal negative guidance.
>
> ### Q6: While NAG is designed explicitly for negative prompting, can the authors extend or generalize the NAG approach to standard (positive-only) text-to-image generation scenarios, without explicitly provided negative prompts?
>
> **Text-to-Image Flux‑Schnell on COCO‑5K**
>
> Good point! NAG can be applied with positive‑only generation simply by using an empty negative prompt (i.e. "" instead of "Low resolution, blurry").
> Even with the empty negative prompt, NAG boosts CLIP and ImageReward, confirming enhanced alignment and perceptual quality.
>
>
> | Flux-Schnell 4-step | CLIP ↑ | FID ↓ | PFID ↓ | ImageReward ↑ |
> |:--------------------:|:---:|:---:|:----:|:-----------:|
> | Baseline            |   31.4  |  25.47   |   38.26   |      1.029       |
> | NAG (negative)      |   32.0 (+0.6)  |  24.46 (-1.01)   |   34.95 (-3.31)   |      1.099 (+0.70)       |
> | NAG (empty)         |   31.8 (+0.4)  |  26.62 (+1.15)   |   37.94 (-0.32)   |      1.083 (+ 0.54)       |
>
> **Class-conditional MeanFlow‑B/4 on ImageNet**
>
> We also evaluate NAG for class‑conditional generation on ImageNet using the single‑step MeanFlow‑B/4 [1] model. By viewing the null class as the negative condition, NAG reduces FID and significantly improves IS.
>
> | MeanFlow-B/4 1-step | FID ↓ | IS ↑ |
> |:--------------------:|:---:|:---:|
> | Baseline            |  13.33   |  135.69   |
> | NAG         |   12.02 (-1.32)  |  155.97 (+20.28)   |
>
> These results confirm that NAG seamlessly supports positive‑only guidance.
>
>
> [1] Mean Flows for One-step Generative Modeling. arXiv 2025
>
> ### Additional Quantitative Metrics
>
> Thanks you for the suggestions. We provide the specialized quantitative metrics HPSv2 and PickScore for few‑step models.
>
> In experiments, HPSV2 exhibits consistent gains with NAG across all models, while PickScore shows mixed results. Across the three primary human‑preference metrics (ImageReward in Table 1, HPSV2, and PickScore), NAG produces reliable improvements in most of them (both ImageReward and HPSV2).
>
> Metrics reported for Baseline / NAG.
>
> |  Arch | Model             | Steps | HPSV2 ↑                   |        PickScore ↑        |
> | :---: | :----------------: | :---: | :------------------------: | :-----------------------: |
> |  DiT  | SANA‑Sprint       |   2   | 27.86 / **28.50** (+0.64) | 22.89 / **22.80** (−0.09) |
> |  DiT  | Flux‑Schnell      |   4   | 30.16 / **31.00** (+0.84) | 22.64 / **22.57** (−0.07) |
> |  DiT  | SD3.5‑Large‑Turbo |   8   | 29.66 / **30.22** (+0.56) | 22.83 / **22.95** (+0.12) |
> | U‑Net | NitroSD‑Realism   |   1   | 29.46 / **29.69** (+0.23) | 22.49 / **22.45** (−0.04) |
> | U‑Net | DMD2‑SDXL         |   4   | 29.85 / **29.98** (+0.11) | 22.83 / **22.82** (−0.01) |
> | U‑Net | SDXL‑Lightning    |   8   | 28.83 / **28.95** (+0.12) | 22.65 / **22.64** (−0.01) |

---

> > ### Comment · Reviewer_oLD6 · 2025-08-04
> >
> > I appreciate the authors' efforts in addressing most of my concerns. However, I remain unconvinced about the theoretical grounding of the extrapolation of the attention space—a point also raised by Reviewer 63rU. While the authors provide a rationale for the attention space of the implicit classifier, there appears to be a logical gap in assuming that the intermediate layer approximates the score function.
> >
> > Specifically, the neural network was trained to fit the output to the score function via a denoising score matching objective. However, this does not inherently guarantee that the intermediate layers, particularly the attention space, exhibit the same behavior or properties. The justification for this extrapolation seems insufficient and lacks theoretical rigor. Without stronger evidence or a more robust theoretical foundation, this claim remains speculative. Given this unresolved issue, I maintain my rating score.

---

> > > ### Author Response · Authors · 2025-08-05
> > >
> > > Thank you again for your feedback. We admit that viewing NAG as an implicit classifier interpolation is heuristic, so below we provide an interpretation from first‐order Taylor approximation.
> > >
> > > ---
> > >
> > > Let $z: c \mapsto G_\theta(x_t, c)$. Its Taylor series around $c$ is
> > >
> > > $z(c+\Delta c) = \sum_{n=0}^{\infty}\frac{1}{n!}\bigl(\Delta c^\top\nabla_c\bigr)^n z(c) = z(c) + \bigl(\Delta c^\top\nabla_c\bigr)z(c) + \tfrac12\bigl(\Delta c^\top\nabla_c\bigr)^2z(c) +\cdots.$
> > >
> > > We consider the first‐order approximation
> > >
> > > $\Delta z \equiv z(c+\Delta c)-z(c) \approx \nabla_c z(c)^\top \Delta c.$
> > >
> > > Next, we can write the network in two stages for a given layer $l$:
> > >
> > > $F_l(c)=G^l_\theta(x_t,c),\qquad z(c)=\bar G^l_\theta\bigl(F_l(c), c\bigr),$
> > >
> > > where $\bar G^l_\theta$ is the tail network after layer $l$. By the chain rule, we have
> > >
> > > $\frac{\partial z}{\partial c}=J_{\bar G^l_\theta}\bigl(F_l(c)\bigr) \frac{\partial F_l(c)}{\partial c} + \frac{\partial \bar G^l_\theta}{\partial c},$
> > >
> > > Where $J$ denote the Jacobian matrix.
> > > By repeatly selecting intermediate layer $l'$ of $\bar G^l_\theta$ and applying chain rule on $\frac{\partial \bar G^l_\theta}{\partial c}$
> > >
> > > $\frac{\partial z}{\partial c}=\sum_l  J_{\bar G^l_\theta}\bigl(F_l(c)\bigr) \frac{\partial F_l(c)}{\partial c}$
> > >
> > >
> > > Applying a first‐order approximation in all $l$, define
> > >
> > > $\Delta F_l \equiv F_l(c+\Delta c)-F_l(c) \approx \frac{\partial F_l}{\partial c}(c) \Delta c.$
> > >
> > > We then obtain a corresponding output‐space increment
> > >
> > > $\Delta z_{feature} \equiv \sum_l J_{\bar G^l_\theta}(F_l) \Delta F_l \approx \nabla_c z(c) \Delta c.$
> > >
> > > Now we have two estimators of the gradient with respect to $c$:
> > >
> > > - $\Delta z$, the output‐space finite difference,
> > > - $\Delta z_{feature}$, the sum of layer‐wise finite differences, each propagated by the Jacobian.
> > >
> > > **CFG** uses the single‐shot estimate $\Delta z$. Gradient ascent gives guidance $\tilde\epsilon = \hat\epsilon + \phi \Delta z.$
> > >
> > > **NAG** instead works in feature space using $\Delta z_{feature}$. At each cross‐attention layer $l$, we inject the per‐layer update $\tilde F_l = F_l + \phi \Delta F_l.$
> > >
> > > In multi‐step sampling, each denoising step is small, so $z(c)$ remains smooth and nearly linear over the condition domain. $\Delta z$ reliably estimates the gradient.
> > > In few‐step regimes, however, the required denoising is too large, causing $z(c)$ to be steeper. $z(c+\Delta c)$ and $z(c)$ no longer lie in a local linear neighborhood, and thus the higher‐order terms in the Taylor expansion are no longer negligible, resulting CFG breaking down.
> > >
> > > On the other hand, a single layer could better preserve linearity under aggressive sampling. By relying on per‐layer $\Delta F_l$, NAG maintains the validity of the Taylor approximation, yielding more stable guidance in few‐step sampling.
> > >
> > > ---
> > >
> > > We acknowledge that, despite the empirical evidence demonstrating NAG’s stability, the Taylor‐based interpretation does not provide a guarantee across all cases. Modern diffusion networks remain “black boxes,” and the behavior of intermediate attention features is still far from fully understood. As such, while the theoretical explanation and experimental results offer a compelling case for feature-space guidance, we recognize that further analysis is required to fully elucidate the mechanisms of NAG’s robustness.

---

> > > > ### Comment · Reviewer_oLD6 · 2025-08-05
> > > >
> > > > Thank you for your kind response. The author has addressed almost all remaining concerns.

---

### Public Comment · ~Wenqi_Guo3 · 2025-11-11
**Some of my thoughts**

This is a great work. The code is well-maintained and easy to use. The main idea is clear, elegant, and easy to follow. It sparked my follow-up work on negative guidance.
Pros:
1. This paper creates an elegant and clean method to introduce negations into non-CFG models; it solves a timely problem, as non-CFG models can dramatically increase generation speed; however, traditional methods lack negation controls, and this method introduces them.
2. It does not require re-training and can be applied on almost every diffusion/flow-matching model, creating a simple plug-and-play method for negative guidance
3. There are a lot of high-quality experiments in the main text and the appendix

I have a few suggestions or concerns for this paper:
1. The authors will benefit from testing the work on a negation setting, e.g., a house with a negative prompt of windows, or a cat with a negative prompt of a specific species. Although the authors did some experiments on this aspect, a quantitative result would be good to see. Authors can take a look for this paper: https://arxiv.org/pdf/2508.10931
2. Although the image quality is improved in metrics like ImageReward and CLIP score, I see minimal visual differences personally. For example, in Figure 5, I can see that the NAG version sometimes has slightly better details than the baseline, but in many cases, the visual improvement is minimal.
3. This work introduced 3 hyperparameters in the generation, and they will affect each other. It might be difficult for users to tune these parameters. Even though the authors suggested a default hyperparameter sets, it might not fit all use cases, especilly in strong negations.

Overall, this paper is good quality. I gained a lot of insights from this paper.

---

### Note · Authors · 2025-08-12

We thank the reviewers for their recognition, constructive feedback, and for leaning towards acceptance after the discussion period.
Strengths of our paper as mentioned by the reviewers:
- "NAG’s core innovation, extrapolating directly in attention feature space, offers rich semantic control over diffusion models, surpassing output-space extrapolation approaches in terms of stability and flexibility" — Reviewer oLD6.
- "Since the method is based on simpler assumptions, this paper shows broad applicability across diffusion model variants, including single-step and video generation tasks." — Reviewer 63rU.
- "NAG adds minimal computational overhead and does not require retraining, making it highly practical for deployment." — Reviewer zEeN.
- "The writing of the paper is clear and well-structured, making the paper easy to follow and understand." — Reviewer P6ev.

During the rebuttal phase, we addressed most of the concerns by:
- Providing a convincing mathematical interpretation for the stability of guidance in feature space.
- Clarifying that NAG operates as a standalone negative guidance in few-step sampling.
- Explaining how NAG preserves manifold integrity.
- Conducting additional ablations and quantitative experiments, showing consistent improvements and the role of normalization.

We will update the paper according to the discussion with the reviewers.

---

### Decision · Program_Chairs · 2025-09-17

**Decision:**

Accept (poster)

**Comment:**

In this work, authors tackle the problem of steering the visual generation process better in adherence to a negative prompt. They propose a method based on attention space manipulation during generation and show its efficacy over CFG. Reviewers were overall in agreement of acceptance of the paper and the discourse between the authors and the reviewers also elicited additional theoretical motivations beyond the intuition and experimental results. The proposed algorithm is intuitive, useful, practical and I recommend acceptance.